



# OMPS LP Version 2.0 Multi-wavelength Aerosol Extinction Coefficient Retrieval Algorithm

Ghassan Taha[1,2], Robert Loughman[3], Tong Zhu[4], Larry Thomason[5], Jayanta Kar[6], Landon Rieger[7], and Adam Bourassa[7]

[1]Universities Space Research Association (USRA), Greenbelt, Maryland, USA
[2]NASA Goddard Space Flight Center, Greenbelt, Maryland, USA
[3]Department of Atmospheric and Planetary Sciences, Hampton University, Hampton, Virginia, USA
[4]Science Systems and Applications Inc., Greenbelt, Maryland, USA
[5]Science Directorate, NASA Langley Research Center, Hampton, Virginia, USA
[6]Science Systems and Applications Inc., Hampton, Virginia, USA
[7]Institute of Space and Atmospheric Studies, University of Saskatchewan, Saskatoon, Saskatchewan, Canada

*Correspondence to*: Ghassan Taha (Ghassan.taha@nasa.gov)

**Abstract.** The OMPS Limb Profiler (LP) instrument is designed to provide high vertical resolution ozone and aerosol profiles from measurements of the scattered solar radiation in the 290-1000 nm spectral range. It collected its first Earth limb measurement on January 10, 2012 and continues to provide daily global measurements of ozone and aerosol profiles from the cloud top up to 60 km and 40 km respectively. The relatively high vertical and spatial sampling allow detection and tracking of sporadic events when aerosol particles are injected into the stratosphere, such as volcanic eruptions or PyroCb events. In this paper we discuss the newly released Version 2.0 OMPS multi-wavelength aerosol extinction coefficient retrieval algorithm. The algorithm now produces aerosol extinction profiles at 510, 600, 674, 745, 869, and 997 nm wavelengths. The OMPS LP Version 2.0 data products are compared to the SAGE III/ISS, OSIRIS and CALIPSO missions and shown to be of good quality and suitable for scientific studies. The comparison shows significant improvements in the OMPS LP retrieval performance in the Southern Hemisphere and at lower altitudes. These improvements arise from use of the longer wavelengths, in contrast with the V1.0 and V1.5 OMPS aerosol retrieval algorithms, which used radiances only at 675 nm and therefore had limited sensitivity in those regions. In particular, the extinction coefficients at 745, 869 and 997 nm are shown to be the most accurate, with relative accuracies and precisions close to 10% and 15% respectively, while the 675 nm relative accuracy and precision are on the order of 20%. The 510 nm extinction coefficient is shown to have limited accuracy in SH and is only recommended for use between 20 - 24 km, and only in the Northern Hemisphere. The V2.0 retrieval algorithm has been applied to the complete set of OMPS LP measurements and the new data set is publicly available.

## 1 Introduction

### 1.1 The importance of stratospheric aerosol measurements

Observations of the stratospheric aerosol layer were first provided by Junge et al. (1961) using balloon measurements, showing a layer extending from 15 to 25 km altitude with a peak at 20 km. Further measurements established that the main composition of the aerosol in this layer is 75% sulfuric acid and 25% water (SPARC, 2006; Deshler, 2008). The stratospheric aerosol budget is dominated by natural sources, where volcanic injections of $SO_2$ and aerosol directly into the lower stratosphere stand out as the largest source over the past decades. Carbonyl sulfide (OCS) makes the second largest contribution to the aerosol layer. These sulfur species originate at the Earth's surface in a variety of reduced forms (including $CS_2$, DMS and $H_2S$) before oxidation in the atmosphere (Kremser et al., 2016). Other sources of stratospheric aerosol include the transport of tropospheric aerosol across the tropical tropopause through large convective systems such as the Asian monsoon (Vernier et al., 2011a; Thomason and Vernier,



2013). In recent years, there is evidence that pyrocumulonimbus (PyroCb) events during large wildfires can inject smoke aerosol into the stratosphere, which can be comparable to volcanic aerosol injections in terms of impact on the stratosphere (Fromm et al., 2010; Peterson et al., 2018; Yu et al. 2019; Torres et al., 2020).

Stratospheric aerosols can have a direct impact on Earth's climate system by affecting its radiative balance, and also play an important role in the chemical and dynamical processes related to ozone destruction in the stratosphere (Hofmann and Solomon, 1989; Solomon et al., 1999; Zhu et al., 2018). Powerful volcanic eruptions such as Mount Pinatubo in 1991 can increase the stratospheric sulfur burden by several orders of magnitude over the pre-eruption levels, which can last for several years and lead to stratospheric warming and tropospheric cooling (Robock, 2000; Deshler, 2008). A volcanic eruption like Pinatubo can also lead to cooling of the surface temperature on the order of few tenths of a degree (Robock and Mao, 1995). It was also shown that the background stratospheric aerosol layer is persistently variable, modulated by weaker volcanic eruptions, thus affecting the global surface temperature (Vernier et al., 2011b; Solomon et al., 2011). Furthermore, Santer et al. (2014) identify statistically significant anti-correlations between observations of stratospheric aerosol optical depth and satellite-based estimates of tropospheric temperature, and show that climate model simulations without the effects of early 21$^{st}$ century volcanic eruptions overestimate the tropospheric warming observed since 1998. Other investigations confirm that early 21st century volcanic eruptions are one of several factors that affect the decreased warming observed during that period (relative to expectations based on continued growth of the greenhouse gases in the atmosphere) (Schmidt et al., 2014; Medhaug et al., 2017). Stratospheric aerosol has also been used as a tracer to study stratospheric dynamics (Trepte and Hitchman, 1992; Jaeger, 2005; Fairlie et al., 2014). Over the past few years, there has been a rising interest in the geoengineering concept for solar radiation management, by injecting aerosol into the stratosphere (Rasch et al., 2008), for which a basic understanding of the present stratospheric aerosol layer is essential. A review paper by Kremser et al. (2016) concluded that it is critical to maintain continuous observational records to detect unpredictable events (like large volcanic eruptions) or unexpected developments (such as non-volcanic processes like strong PyroCb events that result in changes in stratospheric aerosol levels), noting that observations are critical for testing the reliability of climate models.

## 1.2 Overview of stratospheric aerosol measurement types

A comprehensive review of the wide variety of stratospheric aerosol measurements is outside the scope of this study. (The interested reader should refer to Sect. 4 of Kremser et al., 2016, for example.) This section will briefly name some key types of measurements that are useful for determining various properties of stratospheric aerosols, and highlight some advantages and disadvantages of each technique. This overview will focus primarily on the space-based remote sensing methods that are discussed further in Sect. 3-5, especially those that are used directly to assess the performance of the OMPS LP aerosol extinction retrieval.

### 1.2.1 In situ and ground-based measurements

In situ measurements mainly provide information about the stratospheric aerosol size distribution, from which moments (such as surface area density and volume density) can be derived. These instruments directly sample stratospheric air, using a balloon-borne or aircraft platform. Examples include the Optical Particle Counters (OPC, Hofmann and Deshler, 1991; Jaeger and Deshler, 2002), the airborne Focused Cavity Aerosol Spectrometer (FCAS, Jonsson et al., 1995), and the Nucleation-Mode Aerosol Size Spectrometer (NMASS, Brock et al., 2000). Size information can be obtained for particles in the size range 0.05 – 10 microns, depending on the particular instrument. In situ measurements can provide higher temporal and spatial sampling than other techniques during a measurement campaign. But the cost and effort associated with such a campaign limits this method to intensive case studies, rather than providing broad temporal and spatial coverage.



Ground-based lidars also make measurements of stratospheric aerosols (Poole and McCormick, 1988; Chouza et al., 2020). In this case, the signal consists of photons back-scattered by stratospheric aerosols, and therefore the measurements provide the back-scattering coefficient of the particles. This is typically converted into an aerosol extinction coefficient, using an assumed "lidar ratio" as the conversion factor. The signal-to-noise ratio of a single lidar measurement generally is low, which requires combination of many individual shots to produce useful retrievals. Ground-based lidars are typically stationary, so they provide information at just one location, but they can provide high temporal sampling.

### 1.2.2 Space-based occultation measurements

The primary space-based method for characterizing stratospheric aerosol is the occultation method, in which photons emitted by a bright source of radiation are transmitted through the atmosphere as the source rises or sets through the atmosphere (as viewed by the instrument). These line of sight transmission profiles are then converted into vertical profiles of extinction coefficient for the various atmospheric constituents, including aerosol extinction coefficient. The transmission measurement is essentially "self-calibrating," and this fact (combined with the high signal-to-noise provided by a bright radiation source) leads to retrievals with high precision and accuracy. However, the locations and frequency of occultation events are entirely determined by the orbit of the instrument, which limits the coverage that can be achieved.

Most occultation measurements for stratospheric aerosol studies use solar occultation (with the sun as the source of photons). Examples of this technique include the Stratospheric Aerosol Measurement (SAM II) (McCormick et al., 1982), the Stratospheric Aerosol and Gas Experiment (SAGE II) (Chu et al., 1989), and SAGE III (Thomason and Taha, 2003), Polar Ozone and Aerosol Measurement ( POAM II and III) (Lumpe et al., 2002), the Halogen Occultation Experiment (HALOE) (Hervig et al., 1996), the Improved Limb Atmospheric Spectrometer (ILAS I and II) (Hayashida et al., 2000; Saitoh et al., 2006) and the Measurement of Aerosol Extinction in the Stratosphere and Troposphere Retrieved by Occultation (MAESTRO, McElroy et al., 2007; Sioris et al., 2010). Aerosol extinction profiles were also measured by the stellar occultation instrument Global Ozone Monitoring by Occultation of Stars (GOMOS) (Vanhellemont, et al., 2010). The particular solar occultation instrument used in this study (the SAGE III instrument on ISS) is described further in Sect. 3.1.

### 1.2.3 Space-based limb scattering measurements

In recent years, several Limb Scattering (LS) measurements have become available, including SAGE III limb (Rault and Taha, 2007), Optical Spectrograph and InfraRed Imager System (OSIRIS) (Llewellyn et al., 2004; Bourassa et al., 2007), SpectroMeter for Atmospheric CHartographY (SCIAMACHY) (Taha et al., 2011; von Savigny et al., 2015; Malinina et al., 2018 ), and OMPS LP (Rault and Loughman, 2013; Gorkavyi et al., 2013). These instruments measure the radiance scattered by the limb of the atmosphere. The strength of the stratospheric aerosol signal in these radiances depends on several aerosol properties, including the aerosol refractive index, shape, size, and number density. These properties affect both the aerosol phase function and aerosol extinction coefficient, both of which influence the aerosol signal present in the measured limb radiance.

This study evaluates the OMPS LP V2.0 aerosol extinction retrieval algorithm, which estimates the aerosol extinction coefficient based on the measured LS radiance as described in Sect. 2.2. The resulting extinction coefficients are compared to the OSIRIS aerosol extinction product, which is described further in Sect. 3.2.

### 1.2.4 Space-based lidar measurements

Space borne backscatter lidar measurements by Cloud-Aerosol Lidar and Infrared Pathfinder Satellite Observation (CALIPSO) are also used to explore the stratospheric aerosol layer (Thomason et al., 2007a; Vernier et al., 2009; Kar et al., 2019). The


description given in Sect. 1.2.1 for ground-based lidars generally applies to the space-borne lidars as well. The main difference is that an orbiting lidar provides the additional advantage of mobility (near global coverage in the case of CALIPSO). The space-borne CALIPSO lidar instrument is used in this study, and is described further in Sect. 3.3.

### 1.2.5 Summary of available measurements

In 2017, accurate solar occultation measurements of stratospheric aerosols resumed after the deployment of SAGE III on International Space Station (ISS) (Cisewski et al., 2014). Combined with the ongoing OSIRIS and CALIPSO missions, we now have coincident stratospheric aerosol measurements from several space-based platforms. The structure of this paper is as follows: In Section 2 we provide a brief description of OMPS LP instrument and V2.0 algorithm changes. The correlative satellite aerosol measurements are described further in Section 3. Section 4 describes the validation methodology. The comparison results are shown in Section 5, followed by conclusions in Section 6.

## 2 OMPS LP measurements and algorithm description

### 2.1 Instrument review

The OMPS LP sensor images the Earth limb by pointing aft along the spacecraft flight path to measure the sunlit portion of the globe without directly observing the sun. The sensor employs 3 vertical slits separated horizontally to provide near global coverage in 3 - 4 days, and more than 7000 profiles a day. The instrument measures limb scattering radiance at the 290 - 1000 nm wavelength range and the 0 - 80 km altitude range. The instrument is installed in a fixed orientation relative to the spacecraft, which flies in a sun-synchronous ascending orbit with 1:30 PM equator crossing time. As a result, the observed scattering angle (SSA) varies along the orbit, where the Northern Hemisphere (NH) observations correspond to forward-scattered solar radiation and the Southern Hemisphere (SH) observations correspond to back-scattered radiation. Therefore, the aerosol scattering signal is much larger in NH than in SH, resulting in a sampling of the aerosol phase function magnitude varying by a factor of 50 over the course of OMPS orbit (Loughman et al., 2018). OMPS LP is scheduled to fly on the NOAA JPSS-2, 3 and 4 satellites, to extend the stratospheric aerosol measurements into the next couple of decades. (These satellite launches are currently targeted for 2022, 2026, and 2031, respectively.)

### 2.2 OMPS LP V2.0 algorithm improvement

The Version 2.0 (V2.0) OMPS LP aerosol extinction retrieval algorithm builds upon the Version 1.0 (V1.0) and Version 1.5 (V1.5) algorithms, which were described in Sect. 4 of Loughman et al. (2018) and Sects. 2-3 of Chen et al. (2018), respectively. We therefore begin by briefly reviewing the V1.0 and V1.5 algorithms in Sect. 2.2.1 and defining the key variables used. This is followed by Sect. 2.2.2, which details the algorithm updates made to produce V2.0.

### 2.2.1 Review of the V1.0 and V1.5 OMPS LP algorithms

Unlike solar occultation, limb scattering retrievals require complex forward model calculations, and the aerosol retrieval in particular requires assumptions of aerosol refractive index and size parameters. Previous versions of the OMPS LP aerosol retrieval algorithm were mainly designed for minimizing the errors of ozone profiles and level 1 radiance diagnostics. The V1.0 and V1.5 algorithms use OMPS LP radiance measurements $I_m(\lambda,h)$ for a range of tangent heights $h$ at a single wavelength ($\lambda$ = 675 nm) to estimate the aerosol extinction profile. This wavelength was selected to be near the Chappuis band that is used to retrieve the ozone profile in the visible region, for which the OMPS LP radiances are best characterized. The algorithm assesses the measurements





by comparison to two analogous sets of calculated radiances, $I_c(\lambda,h)$ and $I_{c0}(\lambda,h)$. These calculated radiance profiles are generated by the GSLS RTM (Loughman et al., 2004) for the same viewing and solar illumination conditions that existed when $I_m(\lambda,h)$ was

measured. The model atmospheres used (described further below) are identical for these two calculations, with one exception: In the case of $I_{c0}(\lambda,h)$, the model atmosphere contains no aerosols, while the $I_c(\lambda,h)$ model atmosphere contains the first-guess aerosol profile.

The model atmosphere consists of static atmospheric temperature and pressure profiles derived from the operational geopotential height product provided by the NASA Global Modeling and Assimilation Office (GMAO). The algorithm uses the (McPeters and

Labow, 2012) ozone and the PRATMO photochemical box model (McLinden et al., 2000) $NO_2$ climatologies to define the model atmosphere. The first-guess aerosol extinction profile $x_0$ is defined based on a single SAGE climatological profile. Aerosols are assumed to consist of spherical liquid sulfate particles with index of refraction $m = 1.448 + 0i$ (Yue and Deepak, 1983; Wang et al., 1996). In the V1.0 algorithm, the aerosol size distribution (ASD) is assumed to be a bi-modal log-normal distribution (Loughman et al., 2018); this was updated to a gamma distribution in V1.5 (Chen et al., 2018). Mie scattering theory is used to

calculate the aerosol phase function based on the assumed ASD and optical properties.

The Earth's surface is modeled as a Lambertian surface (for which a fraction $R$ of the incident downward radiation is reflected as isotropic, unpolarized upward radiance field at each point). The value of $R$ is determined by requiring that $I_{c0}(\lambda,h) = I_m(\lambda,h)$ at $h = 40.5$ km (Loughman et al., 2018). An approximate ozone correction is also applied to the model radiances to correct for possible ozone error, as described in Sect. 4.3 of Loughman et al. (2018). To reduce the sensitivity of the algorithm to a variety of interfering

factors, the radiances are normalized with respect to tangent height $h$. The measured altitude-normalized radiance (ANR) is defined as $\rho_m(\lambda,h) = I_m(\lambda,h) / I_m(\lambda,h_n)$, with $h_n$ = the normalization tangent height = 40.5 km in the V1.0 and V1.5 algorithms. The value of $h_n$ is generally selected as a compromise between two competing interests: It should be as high as possible (to minimize the atmospheric aerosol extinction at $h_n$), but not so high that the radiance at $h_n$ is poorly characterized (due to residual stray light contamination, low signal-to-noise ratio, etc.). Similar expressions define $\rho_c(\lambda,h)$ and $\rho_{c0}(\lambda,h)$, respectively, based on the calculated

radiance profiles.

As a final step, the ANR values are combined to produce the aerosol scattering index (ASI), which serves as the measurement vector $y$ in the retrieval. The measured ASI is defined as $y_m(\lambda,h) = [\rho_m(\lambda,h) - \rho_{c0}(\lambda,h)] / \rho_m(\lambda,h)$, with similar definitions for $y_c(\lambda,h)$ and $y_{c0}(\lambda,h)$. Since the V1.0 and V1.5 algorithms use a single wavelength, this notation can be abbreviated to $y_m(\lambda,h_i) = y_i^m$, with a similar abbreviation $y_c(\lambda,h_i) = y_i^n$ used to represent the ASI calculated based on the model atmosphere after $n$ iterations of the

retrieval algorithm.

The V1.0 and V1.5 algorithms use the Chahine nonlinear relaxation method (Chahine, 1970) to derive the aerosol extinction coefficient (which represents the state vector $x$) based on the measurement vector $y$ defined above. The state vector is updated iteratively as shown in Eq. (1):

$$x_i^{n+1} = x_i^n \frac{y_i^m}{y_i^n} = x_i^n f_i^n, \tag{1}$$

In this expression, $x_i^{n+1}$ is the state vector at altitude $z_i$ and $n+1$ is the number of iterations. The measurement vector $y_i^m$ is the ASI

at tangent height $h = z_i$, while $y_i^n$ is the calculated ASI based on the extinction profile corresponding to the $nth$ iteration. As shown in Eq. 1, the retrieval creates the updated aerosol extinction coefficient estimate $x_i^{n+1}$ by multiplying the previous estimate $x_i^n$ by the factor $f_i^n$. The V1.0 algorithm constrains the value of $f_i^n$ to lie between 0.2 and 2.0, and sets the number of iterations to $N = 3$. These constraints were primarily motivated by caution in the early stages of developing the aerosol extinction retrieval


algorithm, when the stability of the retrieval was relatively untested. The V1.5 algorithm relaxed these constraints somewhat,

using $N = 4$ and allowing $f_i^n$ values between 1/3 and 3.0.

### 2.2.2 Updates made for the V2.0 OMPS LP algorithm

Since the limb scattering radiances at visible and near-infrared wavelengths are very sensitive to aerosol properties, the V2.0 OMPS LP aerosol algorithm is modified to include multiple wavelengths in this spectral region, similar to the SAGE III aerosol channels (Thomason and Taha, 2003). The V2.0 algorithm uses OMPS LP measurements at wavelengths 510, 600, 675, 745, 869, and 997

nm, selected to minimize the effect of gaseous absorption, with the exception of 600 nm, which will be used primarily for diagnostics. Each wavelength is retrieved independently, as described in the preceding section leading to Eq. 1. Taha et al., (2010) showed that, because of its strong weighting function or Jacobian matrix, retrieving aerosol profiles at longer wavelengths can improve the quality of the profile in the southern hemisphere, where OMPS LP observes backscattered radiation, and extend the retrieval further down in altitude. The Jacobian matrix quantifies the changes in the radiance with respect to the aerosol extinction.

Multiple wavelength aerosol measurements can also provide limited information about aerosol particle size and can be used to identify cloud presence. Notice that the 997 nm radiance measurements are only available after 26 November 2013.

The assumed ASD is the same in V2.0 as in V1.5, but the single first-guess aerosol extinction profile has been replaced by a first-guess climatology that varies with wavelength, latitude, and season, again based on the SAGE aerosol data record. The V2.0 algorithm further relaxes the constraints that were previously applied to the Chahine iteration results: $N = 5$ and $f_i^n$ has an upper

bound of 10.0 and no lower bound. The V2.0 algorithm also checks for convergence after each iteration, rather than always performing the stated number of iterations: Iterations end when the retrieved aerosol extinction changes by < 2% at 20 km.

Limb-scatter instruments such as OSIRIS, SCIAMACHY and OMPS LP suffer from increased stray light at increasing wavelength and altitude due in part to diminishing scattered signal (Jaross et al., 2014; Rieger et al., 2019). To reduce the stray light effect on the retrieval at longer wavelengths, $h_n$ was lowered to 38.5 km in V2.0 (from the 40.5 km value used in previous versions). The

GSLS radiative transfer model used in the V2.0 algorithm was also updated as described by Loughman et al. (2015). The main improvement associated with this change involves use of several zeniths to calculate the multiple scattering source function along the limb line of sight, which improves the radiance calculations near the terminator. Unlike the V1.0 and V1.5 algorithms, the V2.0 GSLS model also includes refraction in the line of sight calculation. The V2.0 algorithm also excludes polarization (which had been included in the V1.5 radiance calculations). The exclusion of polarization is primarily done for speed purposes: Scalar

(unpolarized) radiance calculations are considerably faster than their vector (polarized) counterparts, and the resulting change in $\rho_c(\lambda,h)$ is very small.

Figure 1 and Figure 2 illustrate the contrasting effect of the scattering angle on the measurement vector and subsequent retrieved aerosol extinction profiles at different wavelengths. Figure 1 shows that at scattering angle 154°, wavelengths shorter than 745 nm have poor sensitivity to aerosol, which limits the accuracy and altitude range of the OMPS LP SH aerosol retrieval. In contrast, the

longer wavelengths show high sensitivity to aerosol, thus improving the retrieval significantly at lower altitudes. Notice that the cloud at 10.5 km is only observed by the longer wavelengths. Figure 2 illustrates the strong sensitivity of all 6 wavelengths to aerosol when the scattering angle is small. In the NH, the OMPS LP measurement vector for all wavelengths is positive for all altitudes, and the aerosol retrieval quality does not vary significantly with wavelength. Notice that all retrieved aerosol wavelengths can detect the cloud near the tropopause.


### 3. Correlative aerosol measurements

**3.1 SAGE III/ISS**

The SAGE series of instruments started with Stratospheric Aerosol Monitor (SAM) in 1975, (SAM II) in 1978 (McCormick et al., 1982), SAGE I in 1979, SAGE II (Chu et al., 1993) in 1984, and SAGE III Meteor 3M (M3M) (Thomason and Taha, 2003) in 2001, spanning over 26 years. On February 19, 2017, SAGE III was launched to the International Space Station (ISS) to resume the SAGE series measurements and provide high-resolution vertical profiles of aerosol extinction at multiple wavelengths, the molecular densities of ozone, nitrogen dioxide, and water vapor, as well as profiles of temperature, pressure, and cloud presence. The aerosol extinction is computed as a residual after accounting for Rayleigh scattering and gaseous absorption, and thus, it makes no prior assumptions of the aerosol size or phase function. However, the technique is limited in coverage and number of profiles, to typically about 30 per day. The SAGE III/ISS retrieval algorithm is essentially the same its predecessor on Meteor 3M platform. The quality of the SAGE III on Meteor 3M aerosol data was evaluated by Thomason and Taha, (2003); Thomason et al. (2007b); and Thomason et al. (2010).  These studies found that the aerosol extinction measurements accuracy and precision are on the order of 10% between 15 to 25 km, with the exception of 601 and 675 nm above 20 km, which exhibit substantial bias that was caused by the ozone clearing. A recent study by Wang et al. (2020) about SAGE III/ISS ozone validation also stated that an error in ozone correction caused an under estimation of the aerosol retrievals at wavelengths near the Chappuis band at altitudes where the aerosol loading is minimal. Thomason et al., (2020) also reported a defect in these wavelengths below 20 km due to an error by the $O_4$ cross section used in V5.1.

**3.2 OSIRIS**

OSIRIS (Optical Spectrograph and InfraRed Imager System) is an instrument that measures vertical profiles of limb scattered sunlight from the upper troposphere into the lower mesosphere. It was launched on February 2001 onboard the Odin satellite and continues to take measurements to the present. The instrument measures ozone, aerosol and $NO_2$ profiles. Initial (V5.07) aerosol retrievals were obtained by combining measurements at 470 and 750 nm, and were reported as aerosol extinction profiles at 750 nm.  Rieger et al., (2014) compared coincident aerosol extinction observations by interpolating the SAGE II 525nm and 1020nm channels to the OSIRIS extinction wavelength of 750 nm. They found mean differences of less than 10% in the tropics to mid-latitudes, with larger biases at higher latitudes and at altitudes outside the main aerosol layer.

More recently, the V7 OSIRIS retrieval was introduced, which combines information from measurements at 470, 675, 750 and 805 nm to produce multi-wavelength aerosol extinction retrievals. The expanded wavelength usage reduces biases caused by measurement geometry, and improves the retrieval coverage and quality in the upper troposphere and lower stratosphere (UT/LS) region.  The V7 algorithm also uses a modified version of the Chen et al. (2016) cloud detection algorithm (for PSC detection and general cloud screening). Rieger et al., (2019) report agreement at the 10% level between SAGE II and the Version 7 OSIRIS retrieval, with exceptions at high altitudes, which exhibit low bias due to sensitivity to stray light and nonzero aerosol in OSIRIS normalization altitudes.  Overall, the V7 product agreement with coincident SAGE data is comparable to the V5.07 performance, while the agreement with the CALIPSO-GOCCP product (Chepfer et al., 2010) is improved relative to V5.07. However, Kovilakam et al. (2020) noted that OSIRIS extinction is consistently higher than SAGE II in the lower stratosphere with difference exceeding 30% near the tropopause when comparing monthly means.





### 3.3 CALIPSO


The spaceborne lidar on CALIPSO which was launched in April 2006, provides global measurements of vertically resolved aerosol and cloud attenuated backscatter coefficients at 532 and 1064 nm (Winker et al., 2010). Significant improvement in calibration in V4 of CALIPSO data products makes it possible to retrieve extinction coefficients in the stratosphere even with limited signal-to-noise ratio. The V1 Level 3 CALIPSO stratospheric aerosol profile product was produced using only the nighttime measurements

and substantial spatial (vertical averaging to 900 m, 5° latitude bins, 20° longitude bins) and temporal (monthly) averaging were applied. A constant lidar ratio (extinction to backscatter ratio) of 50 sr was used to retrieve the extinction profiles. The extinction profiles were retrieved using two different methods. In the "background" mode, all detected cloud and aerosol layers were removed and thin cirrus clouds within a few kms above the tropopause were filtered using a threshold on the volume depolarization ratio. In the "all aerosol mode", all layers detected as aerosols in the stratosphere were retained and thin cirrus were filtered using a

threshold on the attenuated color ratio. In this work, we use the gridded extinction profiles from the "all aerosol" mode for consistently comparing with OMPS. It should be noted that in this mode, the cirrus cloud removal is not as efficient as in the "background" mode.

Initial validation of V1.0 CALIPSO L3 532 nm stratospheric aerosol profiles is described by Kar et al. (2019). This study concluded that CALIPSO agrees well with SAGE III/ISS aerosol, with CALIPSO about 25% higher between 20-30km in tropics, and larger

differences at middle to high latitudes and low altitudes.

### 4. Data Comparison Methodology

In order to evaluate the accuracy of OMPS LP aerosol V2.0 retrievals, we have used a variety of methods. This includes comparison with the space-based instruments SAGE III/ISS, OSIRIS and CALIPSO, as well as performing internal consistency tests, which can quantify the uncertainty of the aerosol model assumptions and the diffuse upwelling radiance effect. To provide detailed

assessment of OMPS performance at different altitudes, latitudes, and time, we use two different approaches; coincident observations comparison, and zonal mean climatology comparison. While the first approach is used to eliminate any geographical and time biases, the latter is proved to be useful for monitoring the health and stability of the instrument and retrieval algorithm under different conditions and periods. However, zonally averaged comparisons can produce large biases following large volcanic eruptions, where the aerosol load is high and spatially inhomogeneous, and therefore coincident comparison is preferred under

these conditions (Rieger et al., 2019). The percent difference is defined as

$$Difference = (OMPS – Reference)/((OMPS + Reference)/2) \times 100 \qquad (2)$$

Where reference is the correlative measurement of aerosol extinction. All correlative aerosol profiles were interpolated to 1 km intervals, matching OMPS LP reported altitudes. Zonal mean climatologies were constructed using monthly mean profiles within 5° or 10° latitude bins.

For all comparisons shown in this paper, the center slit aerosol retrieval is used, since it has the most accurate radiometric

calibration and stray light corrections (Jaross et al., 2014). The OMPS LP algorithm identifies cloud top height using the cloud detection method described in Chen et al. (2016). However, this algorithm also flags aerosols from fresh volcanic eruptions or PyroCbs. OMPS LP V2.0 data files now contain both cloud filtered and unfiltered data, as well as separate fields containing cloud height and type. Users may wish to use both cloud height and cloud type flags to filter the data based on their own needs. To avoid removing aerosols from fresh volcanic or PyroCb plumes, we filtered the data by removing the extinction coefficient at and below

cloud top height only if the reported cloud top height is in the troposphere. SAGE III is filtered using wavelength color ratio (Thomason and Vernier, 2013), while OSIRIS and CALIPSO provide cloud screened data.



## 5. Results

### 5.1 Algorithm internal consistency

So as to estimate the uncertainty of the assumed aerosol size model (ASD) and phase function, we compare measurements taken
at similar location but with different viewing geometry. Such measurements take place at high latitudes during the summer of both hemispheres, when the OMPS orbit allows observations of a given latitude in both the ascending and descending nodes. The ascending and descending nodes provide two daily observations of the same latitude, but with different scattering angles. Herein, we compare the daily zonal mean aerosol climatology between ascending and descending nodes in the Northern Hemisphere, where the aerosol signal is stronger.

Figure 3 is a scatter plot of the difference between ascending and descending zonal mean aerosol extinction coefficient in the Northern Hemisphere between 60ºN – 90ºN, plotted as a function of the difference of the single scattering angle (SSA) of the ascending and descending nodes at three different altitudes. The figure shows that at 20.5 km, the V2.0 algorithm has very little, if any sensitivity to the aerosol model errors. At 16.5 km, the dependency of the aerosol retrieval on the scattering angle shows a linear trend of ~0.25% per degree for the 745 and 869 nm, and 0.5% per degree for the 675 nm. The trend is almost doubled to
negative 1% at 25.5 km, although it's distorted by sensor noise and inhomogeneity of the aerosol loading above the *Junge* layer in the northern hemisphere high latitude, especially when events occur inside the polar vortex where the aerosol extinction is very low (Thomason and Poole, 1993). Nevertheless, the increase of the difference per unit of difference in SSA suggests that the aerosol model used in the retrieval is less representative of the aerosol measured at this altitude.

### 5.2 The Diffuse Upwelling Radiance (DUR) uncertainties

As described in Sect. 2.2.1, the aerosol retrieval algorithm uses a simple Lambertian model of the reflecting surface to estimate an effective scene reflectivity ($R$). Although the sensitivity of the aerosol retrieval to Diffuse Upwelling Radiance (DUR) uncertainties is reduced significantly by using normalized radiances (Flittner et al., 2000; Loughman et al., 2018), the error associated with assuming the Lambertian surface is difficult to estimate, and possibly not negligible. In order to quantify DUR uncertainties, we compare OMPS LP daily zonal mean climatology for $R$ values less than 0.3 (cloud free) and greater than 0.3 (bright or cloudy).
Figure 4 is a plot of the percent difference between the two aerosol climatologies at three different wavelengths. The three figures show a very similar picture; very large differences below the tropopause in the tropics, where the cirrus clouds are more frequent, and 5% positive bias above 20 km, just over that cloudy region. The 5% bias may be caused by scattered light originating from cirrus clouds near the tropopause, which wasn't properly accounted for in the radiative transfer model (which simply used a bright Lambertian surface at sea level). The bias is negligible away from the cloudy regions, except for the SH lower altitudes (745 and
869 nm) and the NH (997 nm), which may be caused by a variety of reasons not related to cloud presence, since the large reflectivity outside the tropics is not necessarily an indication of cloud presence. (Larger $R$ values are generally inferred from OMPS LP data at higher latitudes in both hemispheres.)

### 5.3 Comparison of OMPS LP with SAGE III/ISS

### 5.3.1 Coincidence comparison

To evaluate the quality of OMPS LP aerosol retrieval with SAGE III, we use a coincidence criteria of same calendar day measurements, $\Delta lat. = \pm 3º$ and $\Delta lon. = \pm 10º$, which is selected to minimize the effect of spatial and temporal differences between the two instruments. Coincidence pairs are averaged over a 10º (Figure 5, 7, and 8) or 40º (Figure 6) latitudes bins covering the first two years of SAGE III/ISS measurements.




Figure 5 depicts the mean OMPS LP and SAGE III/ISS aerosol extinction profiles for the set of coincident measurements, binned in 10º latitude bins, shown at selected altitudes for 4 different wavelengths. In general, OMPS LP and SAGE III extinction values show similar latitudinal distribution and are well within 20% of each other for most altitudes, with 869 nm showing the best agreement of better than 10%. At 18.5 km in the SH tropics, OMPS aerosol is systemically higher than SAGE III, mostly influenced by cloud presence at lower altitudes. The OMPS 675 nm extinction shows negative bias at 18.5 km that increases with increasing latitudes in SH, where OMPS LP observes the backscatter solar radiation and the attenuation of Rayleigh radiation is substantial below 20 km at 675 nm.

Figure 6 is a summary plot of the mean difference between OMPS and SAGE III coincidences for wavelengths 510, 600, 675, 745, 869, and 997 nm. In general, wavelength 869 nm is the best OMPS retrieved wavelength compared to SAGE with differences less 5% for most altitudes and latitudes, while the other wavelengths agree with SAGE III to within 10%. Exceptions to this occur at high altitudes (above ~28 km) where the aerosol loading is minimal, and near the tropopause, caused by cloud contamination. The


510 and 600 nm OMPS extinction values have a slightly larger bias of 20% in the tropics. In contrast to other wavelengths observed differences, the 997 nm OMPS extinction values have systematic bias of -10% between 60ºS and 20ºN, which might be affected by stray light contamination in the OMPS measurements. Unlike the other wavelengths, the 997 nm laboratory characterization is poor, and its stray light correction therefore has lower quality (Jaross et al., 2014). In the SH, wavelengths 510 nm shows larger bias below 18km, caused by its lack of sensitivity toward aerosol at large scattering angle.


It is worth noting that the best agreement between OMPS and SAGE are found in the NH, where OMPS is observing in forward scattering and the weighting function is strong for all wavelengths. In that region, the agreement is mostly within 5% for altitude between 14 and 22 km. Above 24 km, the observed biases for 510, 600, 675, and to some extent, 745, 869, and 997 nm, gradually increase with altitude, mainly caused by instrument noise and errors under low aerosol conditions, although OMPS assumed aerosol size model uncertainty also contributes to the larger differences.


Figure 7 summarizes the quality of OMPS LP aerosol extinction at 6 retrieved wavelengths, showing the zonally averaged mean differences between OMPS LP and SAGE III aerosol (in percent) at 510, 600, 675, 745, 869, and 997 nm. The comparison shows that below 25 km, the differences between OMPS LP and SAGE III are largely driven by OMPS weighing functions or Jacobians. The weaker Jacobians for short wavelengths under backscatter conditions in the SH and below 20 km leads to limited accuracy, while stronger Jacobians at the longer wavelengths improve its accuracy significantly (Taha et al., 2011; Rieger et al., 2019).


Overall, the shorter wavelengths (510, 600, and 675 nm) are biased low against SAGE III with difference greater than 25% below 20 km in the SH. In addition, these short wavelengths exhibit pronounced large aerosol in the tropics below 20 km caused by the algorithm's reduced accuracy when the measurement vector is very small. The agreement is well within 25% at altitude range 20 - 25 km and better in the NH. Above 25 km, the comparison between the two instruments is poor, caused by either SAGE III ozone correction errors and/or OMPS reduced sensitivity of aerosol at these short wavelengths. The best agreement between OMPS and


SAGE III can be seen at 869 and 997 nm, where they are mostly within 10% of each other for all altitudes and latitudes. The 745 nm OMPS extinction agrees with SAGE to within 15% everywhere except for the SH tropics below 18km.

The standard deviation shown in Figure 8 is influenced by several factors: OMPS LP uncertainties such as measurement noise, forward model errors, and retrieval algorithm sensitivities, in addition to SAGE III/ISS own uncertainty and atmospheric variability. In general, the standard deviation is 15% - 20% for altitudes that show good agreement with SAGE III (Figure 7). The


large standard deviation of ~50% at high altitude is due to instrument noise and low aerosol loading. Below 20 km, the standard deviations for the shorter wavelengths increase to 50%, caused by the OMPS LP reduced accuracy. In the UT/LS, the standard deviation is >50% due to larger dynamical variability, especially during periods when dispersal of plumes due to volcanic eruptions and other events cause longitudinal variations, as well as cloud interference.





Based on SAGE III comparison, we can estimate the OMPS aerosol retrieval relative accuracy to be 10% for 745, 869 and 997 nm in the stratosphere, and 20% for the 675 nm above 20 km and in the NH. The 510 and 600 nm retrievals have limited accuracy in the SH and 25% relative accuracy at altitudes between 20 – 26 km and in the NH. Furthermore, the standard deviation can be used to determine the retrieval relative precision, which can be estimated to be better than 15% for the longer wavelengths, and close to 20% for wavelengths less than 745 nm. The real precision is probably better than the quoted values, since the calculated standard deviation includes atmospheric variability and both instruments biases, none of which was removed (Rault and Taha, 2007; Wang
et al., 2020).

### 5.3.2 Zonal mean comparison

In order to investigate the OMPS LP retrieval performance under different seasonal or geographical conditions, we compare the OMPS LP monthly zonal mean time series with the SAGE III/ISS monthly zonal mean time series for 4 wavelengths at 3 different altitudes. The comparison is also divided into 3 different regions, SH (Figure 9), tropics (Figure 10), and NH (Figure 11). In
general, the agreement between the two instruments in the SH is mostly within 10-20%. The 675 nm extinction at 20.5 km is a notable exception, as the OMPS LP aerosol extinction values drop significantly when the SSA is greater than 145º and the attenuation of Rayleigh scattering below 20 km becomes significant. This behavior appears as an apparent seasonal pattern, in which the OMPS LP / SAGE III difference becomes much larger during SH winter months. It is therefore recommended that OMPS aerosol measurements at λ ≤ 675 nm should be excluded when SSA is greater than 145º below 21 km.
Similar agreement is found in the tropics, at or above 20.5 km, with the exception of the first few months following Aoba volcanic eruption in July 2018, where OMPS LP initially measured more aerosol than SAGE III. This might be caused by the different coverage and frequency of measurements for each instrument, where the monthly zonal mean is heavily skewed by few daily measurements in the case of SAGE III. Still, the difference between the two measurements was mostly within 20% in the aftermath of this volcanic eruption, due in part to OMPS use of fixed background aerosol size distribution model. At 18.5 km, the difference
is often greater than 20%, reaching more than 60% following the subsidence of the volcanic plume. The reason for such large differences is unclear as OMPS LP still shows elevated aerosol levels when SAGE III measurements indicate that the aerosol values are back to pre-eruption levels, although spatial variability and spatial resolutions can contribute to such large differences. SAGE III aerosol extinction profiles are produced on 0.5 km grid with an estimated vertical resolution is 0.7 km (Thomason et al., 2010; SAGE III ATBD, 2002) while OMPS LP vertical sampling is 1.0 km with instantaneous resolution of 1.5 km (Jaross et al.,
2014). Another possible explanation is that OMPS LP cloud clearing can be incomplete and residual cloud contamination can contribute to the large differences near the tropopause. The best agreement between the two instruments can be found at 25.5 km, well within 10%.
In the NH, all OMPS LP wavelengths show similar robust agreement to SAGE III, mostly within 10%, since OMPS LP observes in the forward scattering and all wavelengths are strongly sensitive to aerosol. A notable exception is the first couple of months of
the August 2017 Canadian PyroCb period and the June 2019 Raikoke eruption, when the aerosol loading was very high and spatially inhomogeneous. Spatial inhomogeneity also caused the large bias after 2018 following the sharp drop in aerosol extinction at 25.5 km. While OMPS assumed ASD model may contribute to the larger differences at 25.5 km, instruments noise and calibration errors are also more significant under low aerosol conditions. OMPS LP 997 nm is affected by stray light contamination at the normalization altitudes in the NH high latitudes, which might explain the negative bias during 2019. On the other hand,
SAGE III ozone corrections uncertainty near the Chappuis band can cause a dip in SAGE III aerosol extinction measurements at 676 nm. In particular, the SAGE III 676 nm values at 25.5 km are either zero or negative during 2019 when the measured aerosol is at its lowest levels in the NH during the short lifetime of ISS SAGE III.





**5.4 Comparison OMPS LP with OSIRIS and CALIPSO**

In this section, we compare OMPS LP aerosol at 510 and 745 nm with V7 OSIRIS at 750 nm, and V1 L3 CALIPSO at both 532 and 745 nm. CALIPSO 532 nm extinction ("all aerosol mode") is converted to 745 nm using Angstrom exponent of 1.9, similar to the Angstrom exponent for the OMPS assumed aerosol model. We also included SAGE III/ISS measurements at 755 nm as an independent reference, since SAGE measurements are widely considered as the most accurate stratospheric aerosol dataset (SPARC, 2006; von Savigny et al., 2015; Kremser et al., 2016; Thomason et al., 2018; Kar et al., 2019). Although coverage and

sampling differences can make such comparisons difficult, it provides a chance to evaluate the entire OMPS LP data record relative to these two datasets. As both OSIRIS and CALIPSO approach the ends of their lives, it is now more critical than ever to extend the stratospheric aerosol record that has been developed from SAGE/OSIRIS/CALIPSO into OMPS LP/SAGE III/ISS records.

Figure 12, 13 and 14 show OMPS LP, SAGE III/ISS, OSIRIS, and CALIPSO zonally averaged monthly mean aerosol extinction coefficient at 3 different altitudes in the SH, Tropics, and NH respectively, spanning a period between 2012 and 2019. Right panel

is the mean difference in percent between OMPS LP and all instruments for the same altitudes and latitudes. In general, OMPS LP, OSIRIS, CALIPSO, and SAGE III aerosol measurements are closely matched at all altitudes, showing enhanced aerosol following the eruptions of Nabro (June 2011), Kelut (February 2014), Calbuco (April 2015), Aoba (July 2018), Raikoke (June 2019), and Ulawun (August 2019), as well as the Canadian fires (August 2017). At 25.5 km in the tropics, the OMPS LP, CALIPSO, and OSIRIS clearly show enhanced aerosol layer within one year of each volcanic eruption, lofted into this altitude by

the upwelling tropical branch of Brewer-Dobson circulation (Vernier et al., 2011b). Notice that agreement between OMPS LP and the 3 instruments is generally within 20% for all shown altitudes, except for the 25.5 km in SH, where CALIPSO is somewhat biased high, and in the tropics at or below 20.5 km, where the aerosol loading is greatly enhanced by several moderate volcanic eruptions. Part of this large difference can be due to aerosol model uncertainties, as both OMPS LP and OSIRIS assume fixed background aerosol model, while CALIPSO uses fixed lidar ratio. In addition, differences at 18.5 km tropics can be affected by

residual cloud contamination, as all three instruments use different criteria for screening cloudy events. Kar et al., (2019) reported that CALIPSO have larger biases 2–3 km above the tropopause that might be due to cloud contamination. The OMPS LP 510 nm comparison shows good agreement with CALIPSO at 20.5 km in the SH, with periods of larger difference when the SSA is greater than 120° in the SH. At 18.5 km, the difference is also 20% with OMPS exhibiting periodic jumps in the aerosol extinction values at 510 nm, caused by the algorithm's reduced accuracy when the measurement vector is very small (see Figure 1). At 25.5 km,

both the 510 and 745 nm extinction values show similar variability to CALIPSO, well within 25%, except for the SH. In the NH, the accuracy of the 510 nm aerosol retrieval is comparable to the 745 nm accuracy. OSIRIS monthly means in the NH are slightly noisier because of the limited number of profiles used.

A summary of the comparison between OMPS LP and CALIPSO is shown in Figure 15(a, b). The differences between OMPS LP and CALIPSO are in general within 25% between 50°S and 50°N, except for the tropics, which is closer to 25% at some altitudes

and greater in case of 510 nm. The difference is substantially large in mid to high latitudes of both hemispheres, with CALIPSO showing rather large bias of 50%, and exceeding 100% at altitudes above 25 km., This is also consistent with an earlier study that shows agreement of 25% between CALIPSO and SAGE III/ISS between 30°S and 30°N, and larger biases of 100 - 200% at middle to high latitudes (Kar et al., 2019). They also noted that the primary parameter affecting the comparison is likely the fixed lidar ratio of 50 sr used in the CALIPSO retrieval, which is dependent on the aerosol optical and physical properties. It is plausible that

the comparison between OMPS and CALIPSO can be further improved by using different lidar ratio.

Agreement between OMPS LP and OSIRIS (Figure 15c) is generally very good, with differences less than 20% for most latitudes in the stratosphere below 30 km, except for the tropics, where OSIRIS is 30% low compared to OMPS. The reason for the increased



differences in the tropics is unclear, however, a similar negative bias of 15% was also noted for comparisons in the tropics between OSIRIS and SAGE III/ISS (Rieger et al., 2019), while OMPS 745 nm has 10% positive bias in the tropics relative to SAGE III/ISS (Figure 5). Combining both biases might explain the difference seen in the tropics. The large bias seen at NH high latitude above 25 km is consistent with comparison with SAGE III and CALIPSO and highlight the difficulties of retrieving very low aerosol for both instruments. Rieger et al. (2018) also reported large negative bias at high latitude above 25 km when comparing OSIRIS to SAGE II that is caused by non-zero aerosol in the normalization altitudes.

Similar to the previous comparison with SAGE III (Figure 7), the standard deviation for the three comparisons is generally less than 15% for altitudes that shown good agreement with either CALIPSO or OSIRIS (Figure 15d, e, and f). There is a large standard deviation of ~50% above 22 km at the SH high latitude due to OMPS reduced accuracy under low aerosol loading and large scattering angle. In the UT/LS, the standard deviation is greater than 40% due to larger dynamical variability, especially during volcanic eruptions, and cloud interference.

### 5.5 Recommendations for use of OMPS LP aerosol extinction data

OMPS LP provides good quality multi-wavelength aerosol extinction retrievals that can be useful for scientific studies. In particular, we find the relative accuracy at 745, 869 and 997 nm is on the order of 10% and relative precision better than 15% in the primary aerosol layer in the stratosphere. These retrievals are suitable for continuing the long-term record of stratospheric aerosol that was started by SAGE II in 1984, although the OMPS 997 nm retrieval is affected by stray light contamination and shows slight negative bias in the SH. Since the 869 and 997 nm retrievals have the strongest sensitivity to aerosol in almost all altitudes and regions, these are most suitable for scientific studies like detection and tracking periodic events when aerosol particles are injected into the stratosphere, such as volcanic eruptions or PyroCbs. The 675 nm relative accuracy is on the order of 20% above 20 km, and its relative precision is 15%. In the SH, its accuracy is reduced, with measurements deemed unusable when the SSA exceeds 145°. We find the 600 nm of comparable accuracy to 675 nm at altitudes 20-25km which suggests that the applied ozone corrections at this wavelength are reasonable. Still, we recommend the user avoid using this channel since it is only meant for diagnostic purposes. The 510 nm relative accuracy is 25% at a limited altitude rage of 20-24 km in the SH and tropics, with similar accuracy in the NH for all altitudes below 25 km. Because of its weak sensitivity to aerosol in the backscatter, we recommend cautious use of the 510 nm retrieval, only in the NH. We also recommend that the user be cautious when attempting to derive aerosol size information from quantities such as Angstrom exponent, since the accuracy of each wavelength retrieval is affected by its weighing function at some altitudes and latitudes, and the 997 nm retrieval is affected by stray light contamination, which may bias the result.

### 6. Conclusions

The new V2.0 OMPS LP aerosol extinction products at wavelengths 510, 600, 675, 745, 869, and 997 nm have been processed. Comparisons with coincident measurements by the SAGE III/ISS, OSIRIS and CALIPSO instruments indicate that the OMPS LP retrievals are suitable for scientific studies. By comparing OMPS measurements at different scattering angles, we demonstrate that the retrieval's dependency on viewing geometry is negligible at 20.5 km and reduced significantly for the longer wavelengths at lower altitudes relative to the short wavelengths. In addition, we estimate the uncertainty in the aerosol retrieval caused by diffuse upwelling radiance (DUR) to be in the order of 5%. The 745, 869 and 997 nm extinction profiles are shown to be the most accurate and most suitable for continuing the long-term record of stratospheric aerosol, with relative accuracies and precisions close to 10% and 15% respectively, while the relative accuracy and precision of 675 nm extinction profiles are on the order of 20%. Differences



can be larger for individual profiles or zonal mean comparison, which can be affected by differences in instrument's coverage and inhomogeneity along the line of sight for fresh volcanic eruptions. The 510 nm extinction profile was shown to have limited accuracy in SH and is only recommended for use between 20-24 km and in the NH only. The 600 nm extinction profile is mainly retrieved for diagnostic purposes and is not recommended for scientific use. Future versions of the OMPS LP retrieval algorithm may improve on the assumed aerosol size model to account for the different types of aerosol at different altitudes. Additionally, a
better cloud clearing which can utilize OMPS multiple wavelength dependence may further improve the aerosol products in the UT/LS region.

**Code and data availability.**

OMPS-NPP LP L2 Aerosol Extinction Vertical Profile swath multi-wavelength daily 3slit Collection 2 V2.0, are accessible from Goddard Earth Sciences Data and Information Services Center (GES DISC), accessed [data access date],
doi:10.5067/CX2B9NW6FI27. SAGE III/ISS data (https://doi.org/10.5067/ISS/SAGEIII/SOLAR_HDF4_L2-V5.1) and CALIPSO data (https://eosweb.larc.nasa.gov/project/calipso/calipso_table) are accessible at the NASA Atmospheric Sciences Data Center. OSIRIS data set can be downloaded from https://arg.usask.ca/docs/osiris_v7/. Data analysis products shown here are available from the corresponding author.

**Author contributions.**

GT and RL were responsible for the development of the OMPS LP V2.0 multi-wavelength algorithm, which is described in this paper. TZ was responsible for code improvements and testing. GT wrote the initial draft of the paper with help from RL. GT carried out the analysis shown here. LT participated in the scientific discussion about SAGE III data. JK participated in the scientific discussion about CALIPSO data. LR and AB participated in the scientific discussion in regard to OSIRIS. ALL authors reviewed the manuscript and provided with advice on the text and figures.

**Competing interests.**

The authors declare that they have no conflict of interest.

**Acknowledgment**

This work is funded by NASA contract 80NSSC18K0847.

The authors would like to thank the OMPA LP characterization team lead by Glen Jaross for producing the Level 1 gridded data,
the OMPS ozone SIPS team lead by Matt Deland and Colin Seftor for Level 2 data production, OSIRIS, CALIPSO and SAGE III-ISS teams for providing the data used in this study.

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





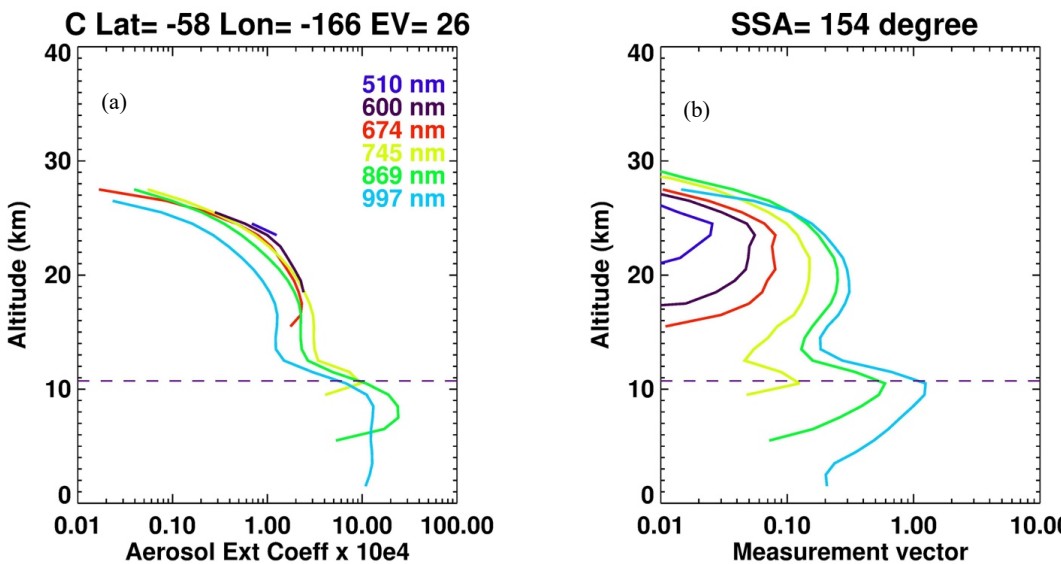

**Figure 1: Plot of retrieved OMPS LP aerosol extinction profiles (x $10^4$ km$^{-1}$) colored by wavelength for event 26 in SH measured on 12 April 2018 (a) and aerosol scattering index (ASI) or measurement vectors (y) for the same event. Dash line is the tropopause altitude.**





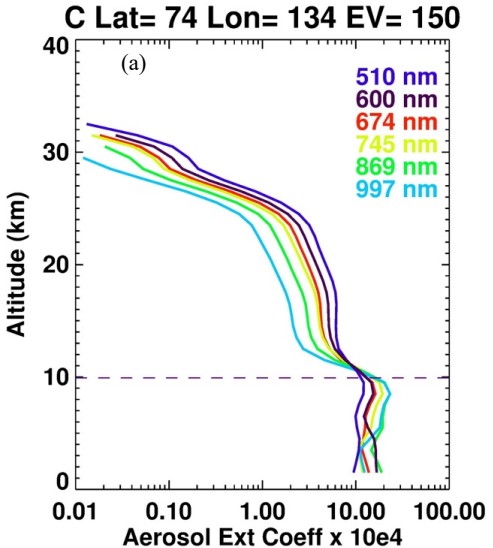

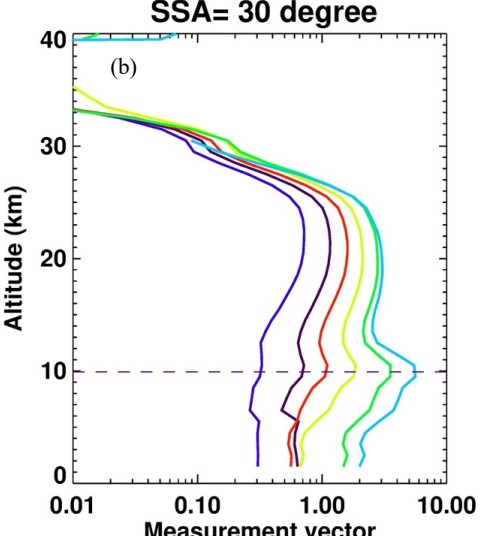

**Figure 2: Same as Figure 1 but for event 150 in NH.**



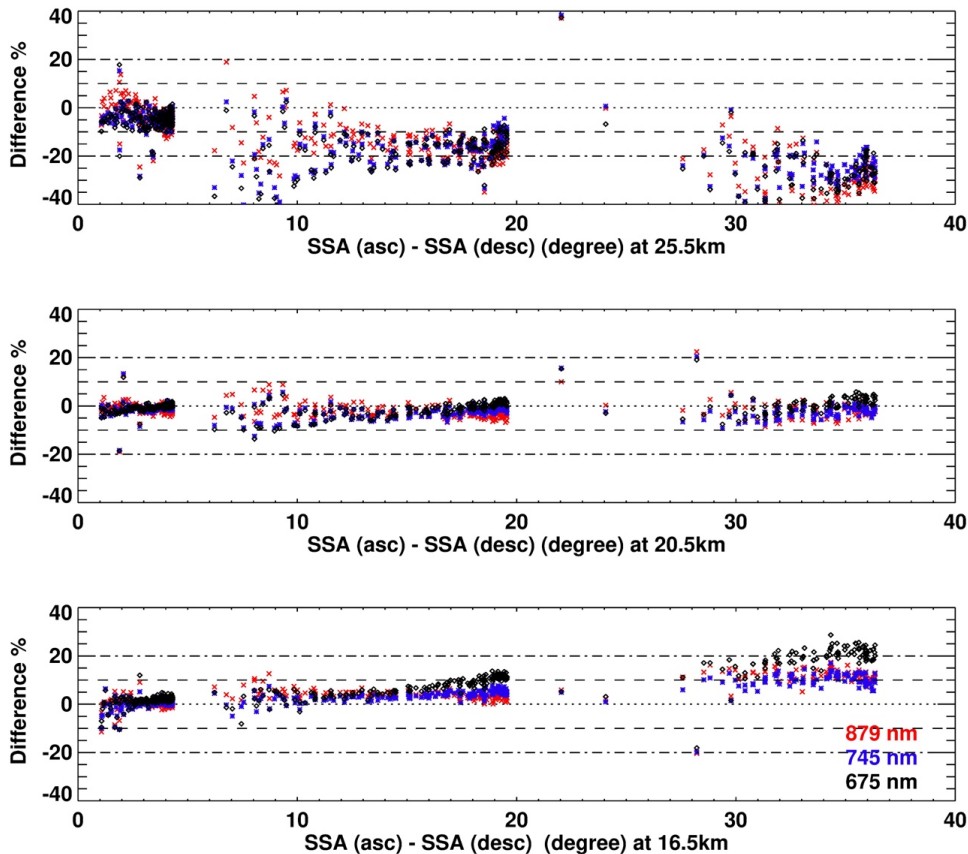

**Figure 3: Scatter plot of the difference between ascending and descending aerosol extinction coefficient zonal means (%) for 675 nm (black), 745 nm (blue) and 869 nm (red) vs. the difference of single scattering angle (SSA) of the ascending and descending measurements at 25 km (top), 20.5 km (middle), and 16.5 km (bottom). Measurements where the SSA difference is less than 6° are for zonal mean latitude (90° N – 80° N), SSA difference (6° – 25°) are for (80° N – 70° N) and SSA difference (25° – 36°) are for (70° N – 60° N).**


**(a)** $\text{Aer}_{gt\,0.3} - \text{Aer}_{lt\,0.3}$ **(745 nm)**   **(b)** $\text{Aer}_{gt\,0.3} - \text{Aer}_{lt\,0.3}$ **(869 nm)**   **(c)** $\text{Aer}_{gt\,0.3} - \text{Aer}_{lt\,0.3}$ **(997 nm)**

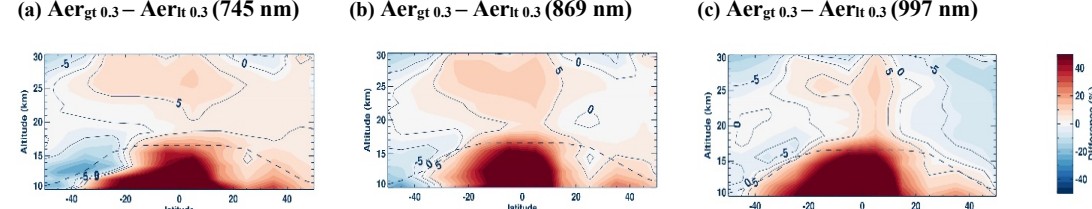

**Figure 4:** Plot of the difference between the aerosol climatology when reflectivity is greater than 0.3, and when reflectivity is less than 0.3. Extinction climatology at 745 nm. (a), 869 nm (b) and 997 nm (c). The dashed line is tropopause altitude. Contour line shows differences greater than $\pm\,5\%$.





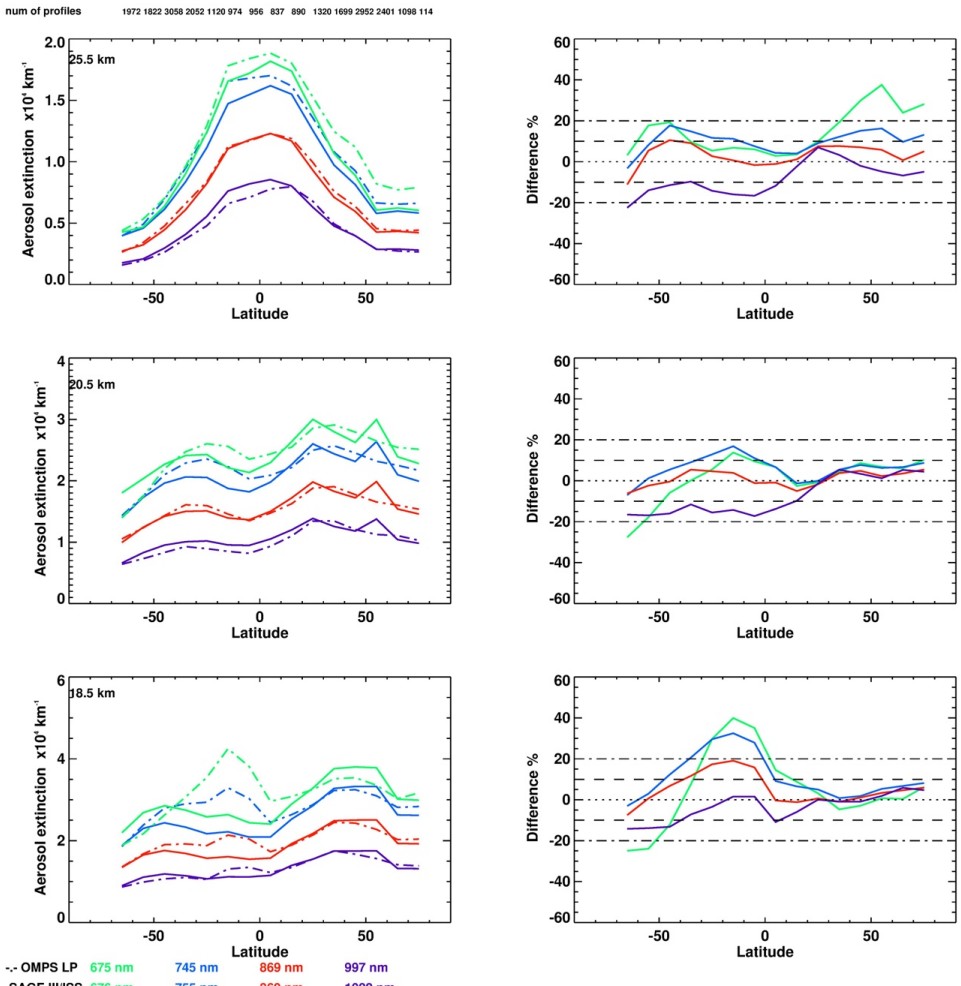

**Figure 5:** Left panel is OMPS v2.0 (dash) and SAGE III (solid) aerosol extinction coefficient (x10⁴ km⁻¹) at 25.5, 20.5 and 18.5 km, for 675 nm (green), 745 nm (blue), 869 nm (red), and 997 nm(violet). Right panel is the percent difference between the two measurements. The number of coincidences for each zone is shown on left top panel.





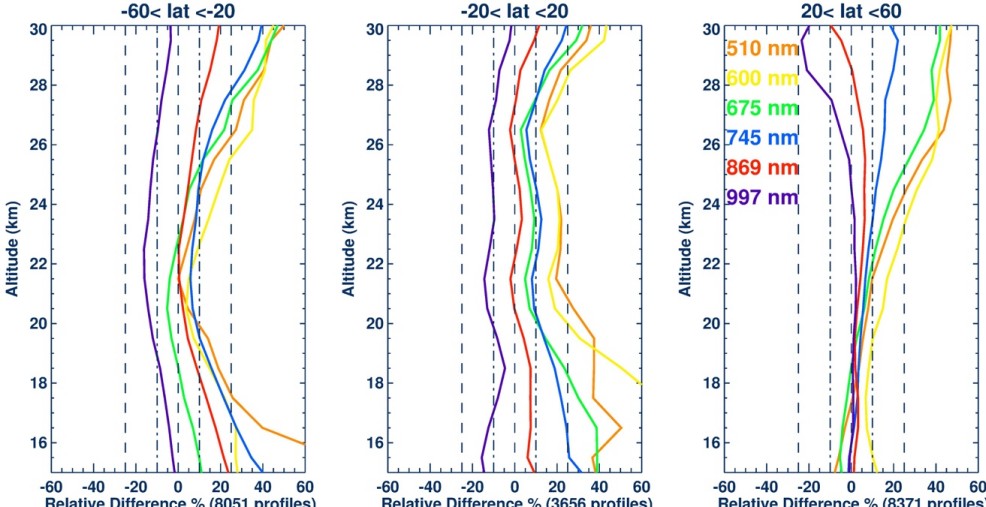

**Figure 6: Summary plot of the average percent difference between OMPS LP and SAGE III profiles in percent at 3 different latitudinal zones, for 6 wavelengths: 510 nm (orange), 600 nm (yellow), 675 nm (green), 745 nm (blue), 869 nm (red), and 997 nm(violet).**





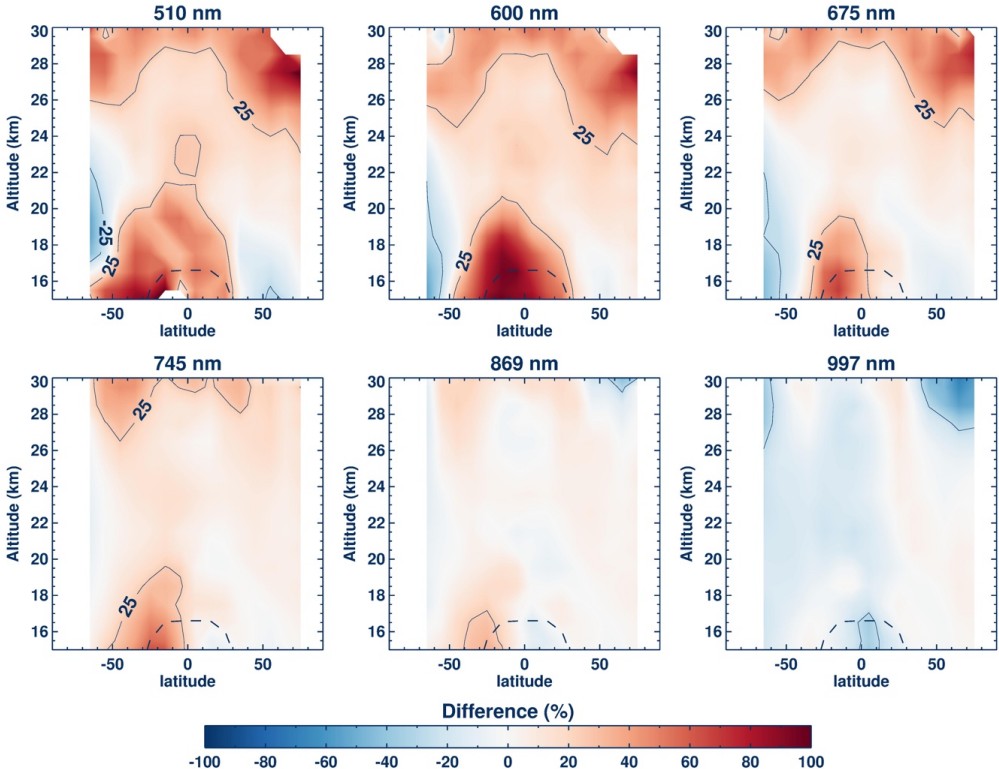

**Figure 7: Mean differences between OMPS LP and SAGE III extinctions as a function of latitude and altitude at 510, 66, 675, 745, 869, and 997 nm, zonally averaged at 10º latitudes. Contour line shows differences greater than ±25% and dash line is the tropopause altitude. Positive differences (in percent) indicate the OMPS LP values are higher than SAGE III/ISS.**



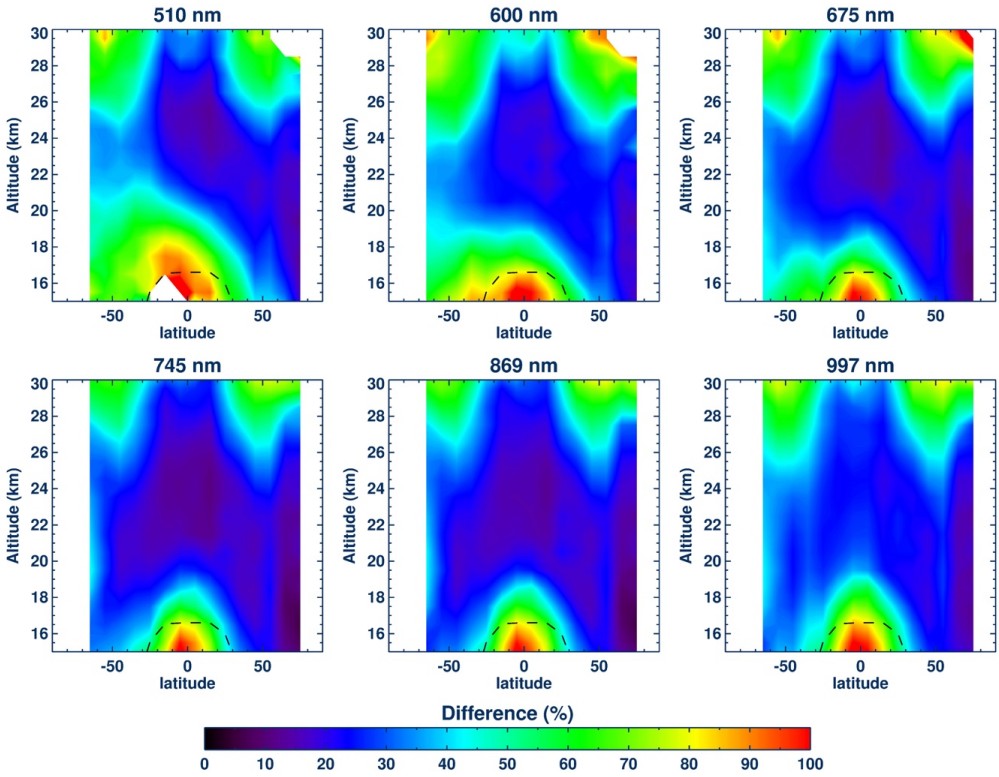

Figure 8: Same as Figure 7 but for the 1 - σ standard deviation or spread of difference between OMPS LP and SAGE III.





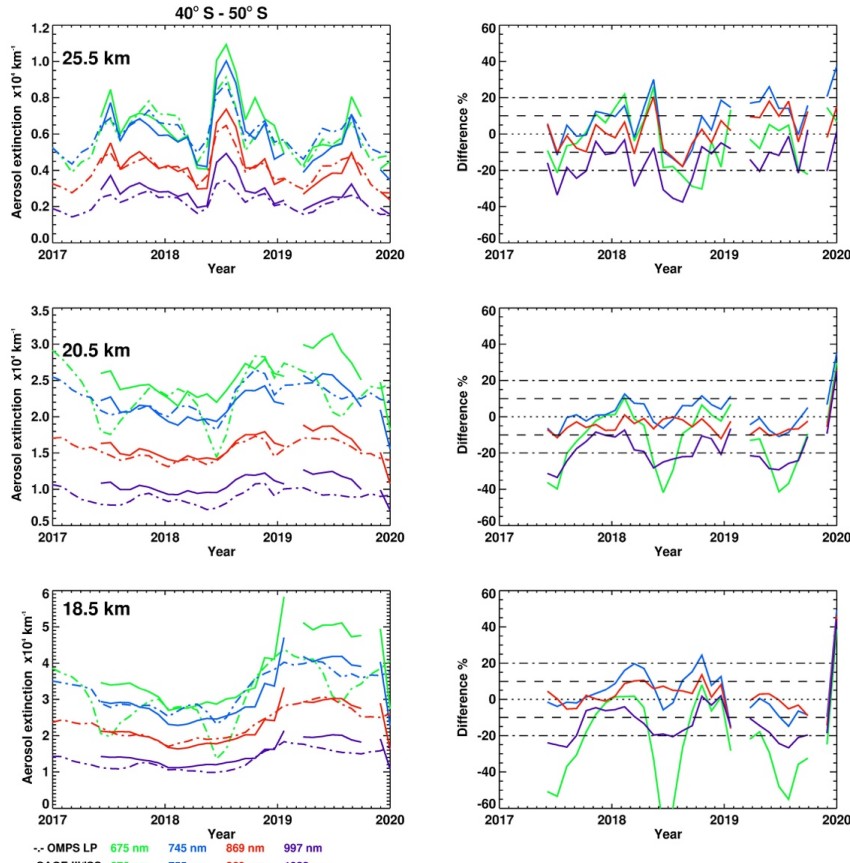

**Figure 9: Left panel is OMPS LP (dash) and SAGE III/ISS (solid) aerosol extinction monthly zonal mean at 675, 745, 869, and 997 nm, latitude zone 40° S - 50° S for altitudes 25.5 km (top), 20.5 km (middle) and 18.5 km (bottom), from 2017 to 2019. Right panel is the percent difference.**





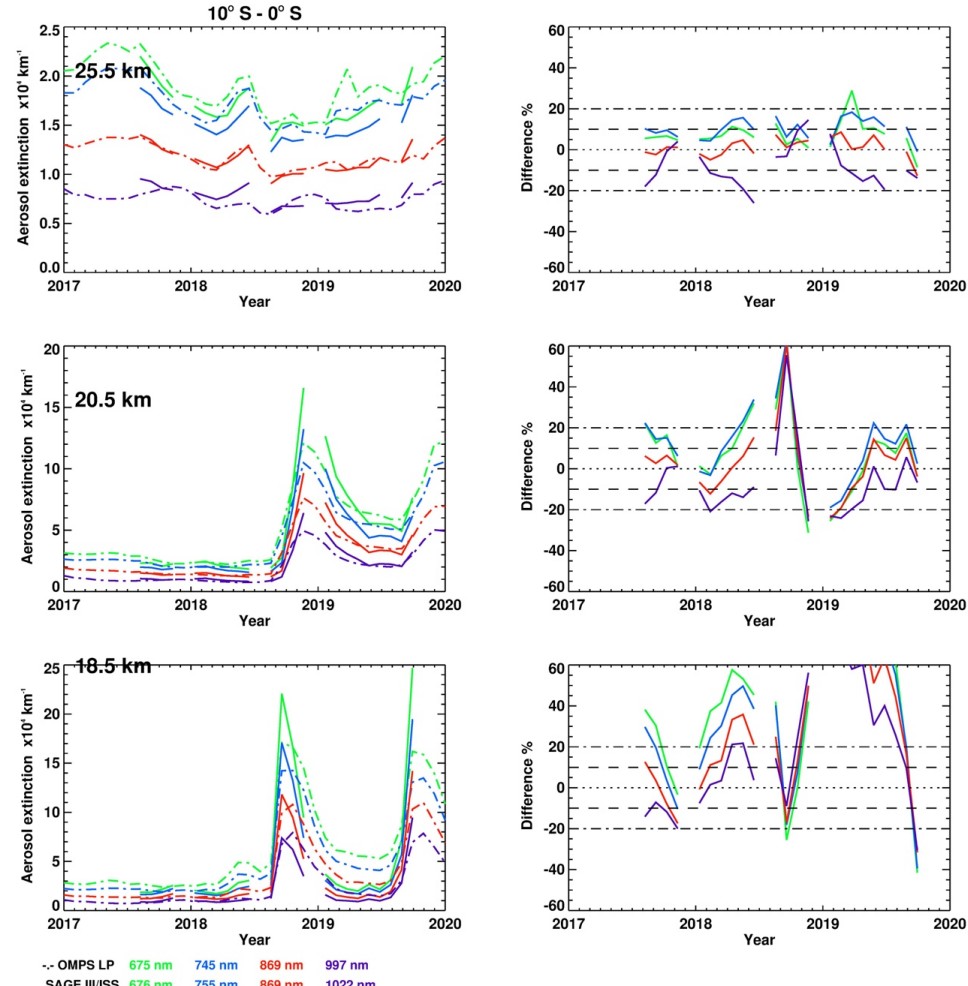

**Figure 10: Same as Figure 9 but for latitude zone 10° S - 0° S**





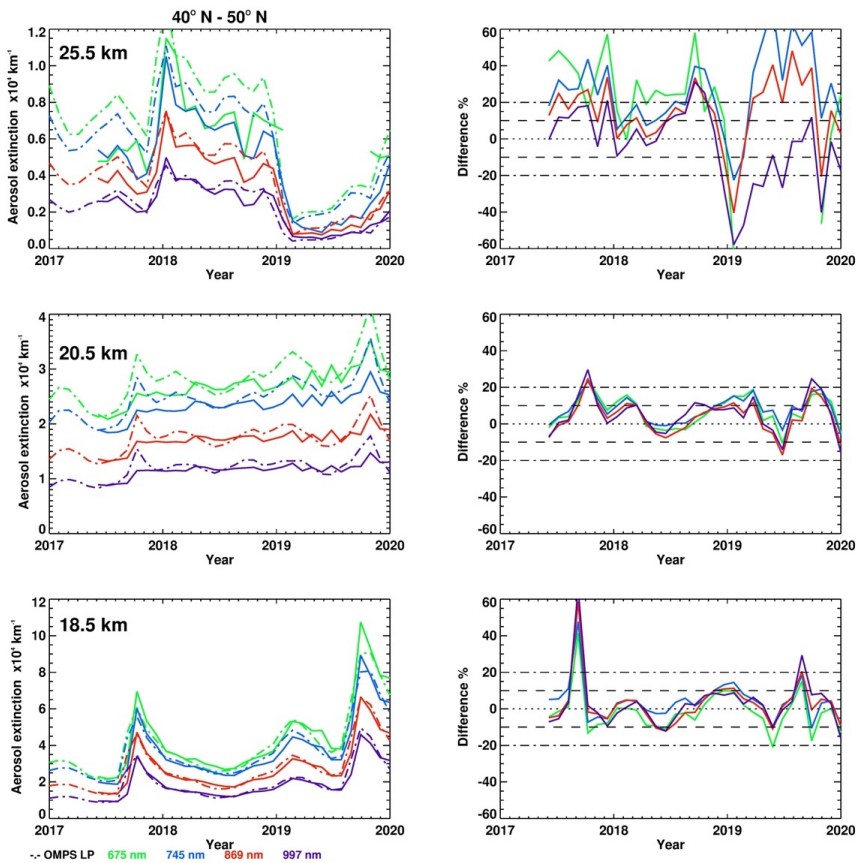

**Figure 11: Same as Figure 9 but for latitude zone 40° N- 50° N**





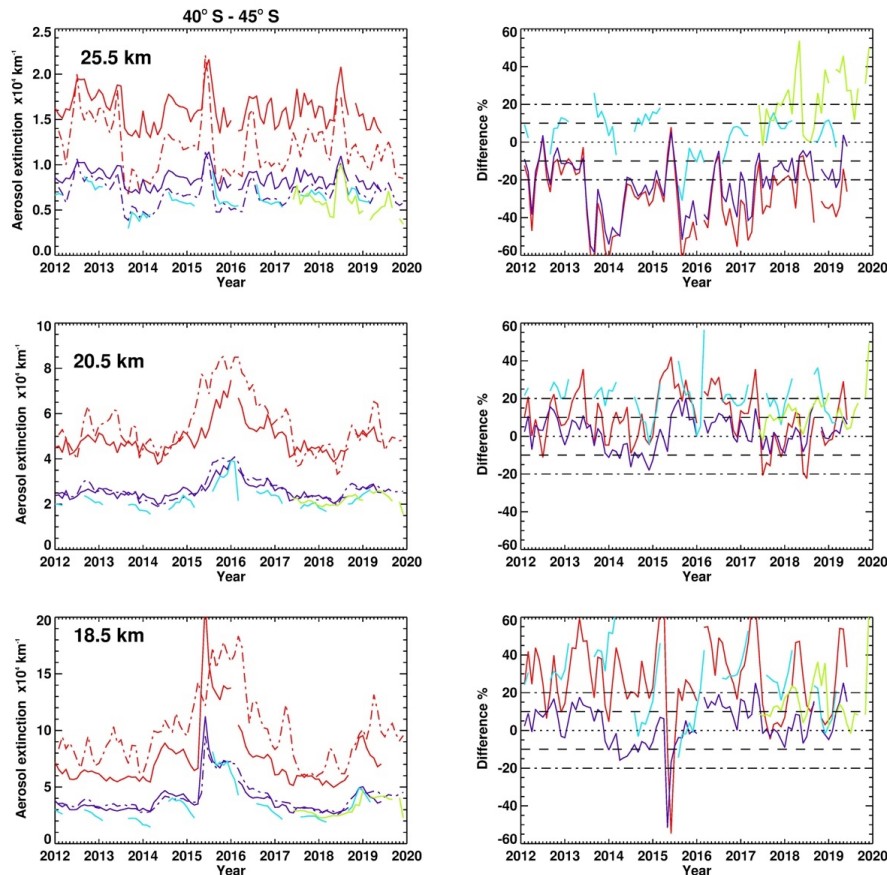

**Figure 12: Left panel is CALIPSO 532 nm (red) and 745 nm (blue),OMPS LP 510 and 745 nm (red and blue dash line), OSIRIS 750 nm (light blue), and SAGE III/ISS 755 nm (green) aerosol extinction monthly zonal mean, latitude zone 40º S - 45º S for altitudes 25.5 km (top), 20.5 km (middle) and 18.5 km (bottom), from 2012 to 2019. Right panel is the percent difference between OMPS LP and other instruments.**





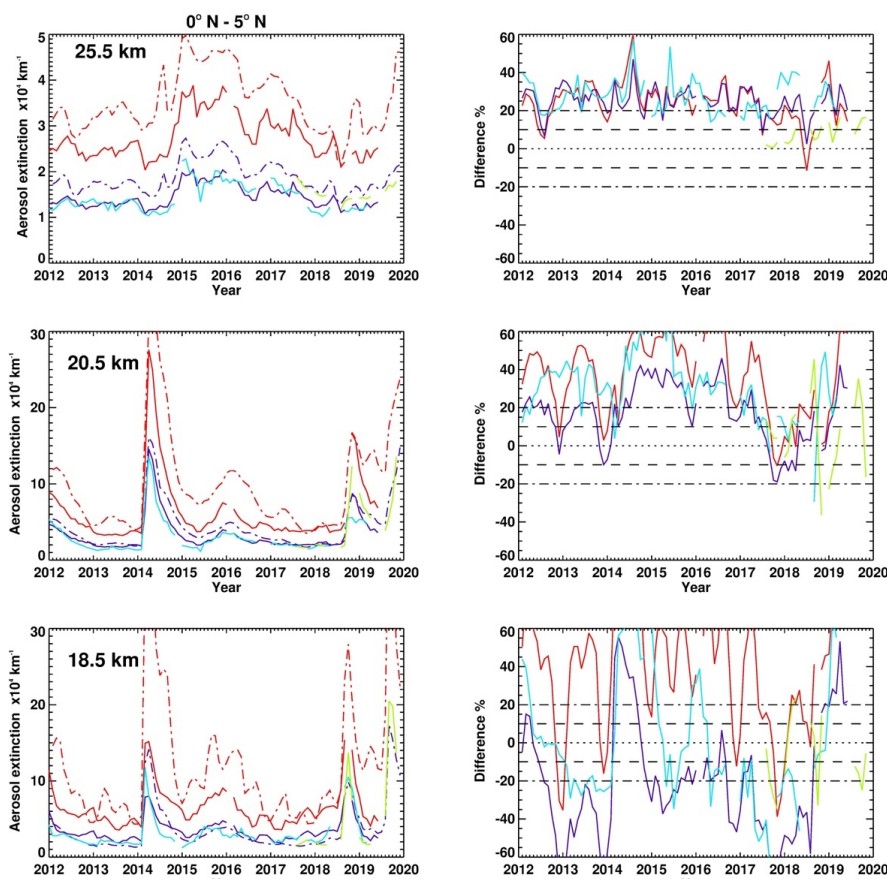

Figure 13: Same as Figure 12 but for 0° and 5° N.





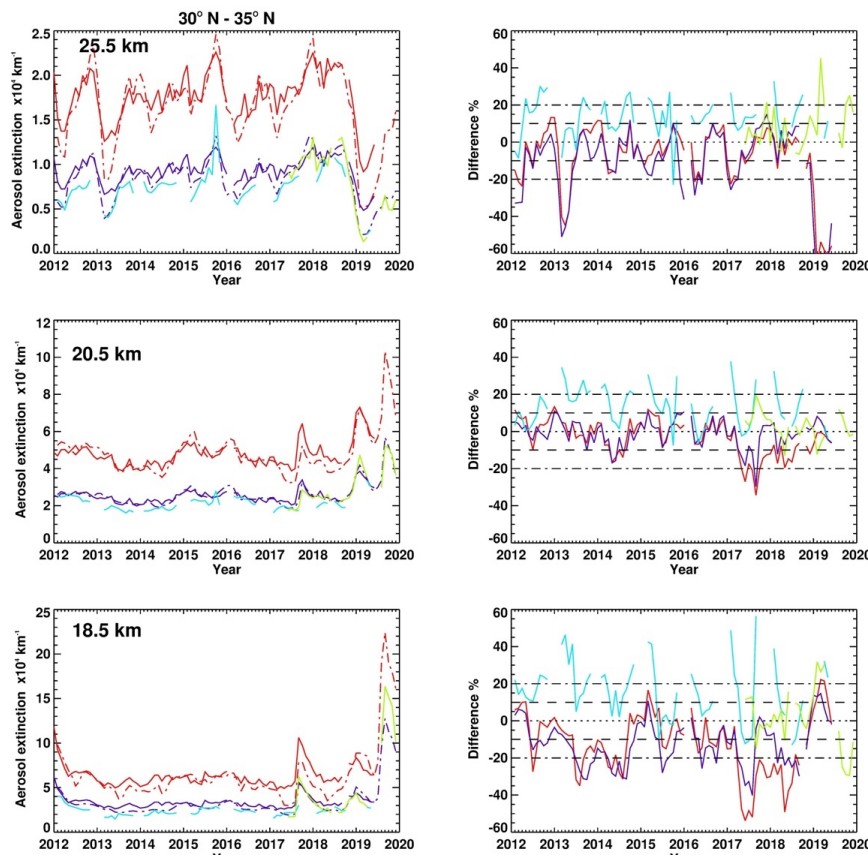

**Figure 14: Same as Figure 12 but for 30° N and 35° N.**





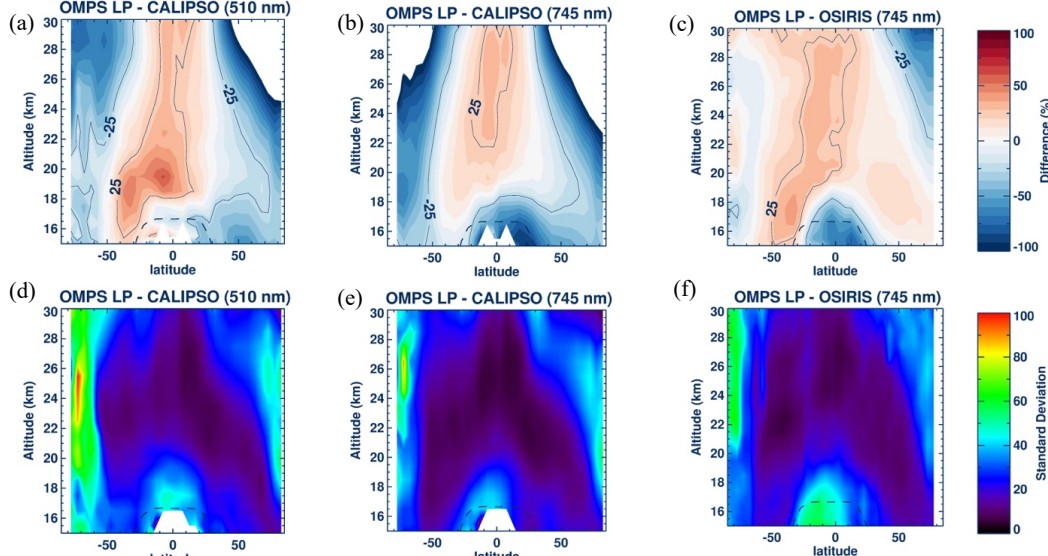

**Figure 15:** Top panel is the mean difference between OMPS LP and (a) CALIPSO 510 nm (b) CALIPSO 745 nm and (c) OSIRIS 745 nm measurements in percent. Dashed line is the tropopause altitude and contour line is for difference greater than ± 25%. Lower panel is same as top panel but for the 1 - σ standard deviation or spread of difference between OMPS LP and CALIPSO 510 nm (d), CALIPSO 745 nm (e), and OSIRIS 745 nm (f).
