# Peer review of "OMPS LP Version 2.0 Multi-wavelength Aerosol Extinction Coefficient Retrieval Algorithm"

_Atmospheric Measurement Techniques, 2020_

## Referee Comment (RC1) · Anonymous Referee #1 · 5 Sep 2020

Review of Taha et al. (submitted to AMT)

I have combined the scientific/major comments with some wording suggestions below. The work is certainly publishable after addressing some of the revisions, but I feel the method can be improved (see below) and wonder if improved results would follow.

L166: "$h_n$ = the normalization tangent height = 40.5 km" -> "$h_n$, namely the normalization tangent height, set to a value of 40.5 km"

L169: Similar -> Analogous

L182-185: This is more constrained, not relaxed, on the low end (0.333 vs 0.2). Is "0.2" a typo?

L201: This is my first major criticism: stopping the retrieval based on convergence at one height is a mistake. This will mean poor low accuracy particularly below this tangent height but even at altitudes at/above this tangent height.

L227: "it" is not defined

L235: "by" -> "caused by"

L236: Sect 3.1 should state something more about SAGE III cloud detection if SAGE III data have been filtered for clouds in this paper. This becomes relevant with the statement at L395.

L258: Regarding "retrieve extinction", CALIPSO does not really retrieve extinction.

L261: retrieve -> obtain

L269: 30km -> 30 km (see also L344, L468)

L281: Where -> where

L288: What is a "cloud type flag" and how is this determined?

L305: "negative 1%" -> "-1% per degree"

L305: Why is "Junge" italicized?

L306: "in the northern hemisphere high latitude" -> "at northern high latitudes"

Figure 3: Why is there a wavelength dependence to this error? The wavelength dependence is obvious at 16.5 km. Are all months included here? Without doing any calculations, it seems that there should be some larger scattering angle differences when the sun is low (winter) for 60-70 degrees N. Why is the sign of the SSA dependence of the aerosol extinction difference changing (y-axis) between 25 and 16 km? I wonder if specifying the convergence criterion only at 20 km (see comment above) is partly responsible for the lack of SSA dependence at 20 km, while there is an obvious dependence at 25 and 16.5 km.

L316: Since at L290, the authors inform us that clouds are being removed, any differences in aerosol extinction between low and high $R$ are therefore not expected differences in extinction due to cirrus, but rather point to cirrus being missed by the cloud flagging. I suppose this is difficult to avoid in the tropical tropopause region.

L322: Do the authors believe that the larger *R* (effective scene reflectivity) at higher latitudes is real or an artifact of the retrieval? If it is real, *R* should have a seasonal dependence, being higher in winter when there is snow covering the land at northern high latitudes.

Figure 4: This figure can be improved since the contours also look like dashed lines in certain spots.

L337: wavelength 869 -> 869

L343: Delete "wavelengths"

L344: Lack of sensitivity does not necessarily result in bias, in should result in larger noise. Can the authors say anything insightful about the bias (e.g. the sign of the bias)?

Figure 6: It seems a bit odd that the aerosol extinction bias relative to SAGE is higher at 600 nm than 510 nm for low latitudes at/below 18 km?

Figure 6: The retrieval clearly has a systematic error in the southern hemisphere. The behavior in the northern mid-latitudes is what one would expect and hope for….

L359: of -> to

L369: 10% is a bit optimistic, particularly at 30 km. Could "~10%" be written instead?

L374: instruments -> instruments'

L388: This sentence is unclear. The authors imply that the difference would be even less than 20% if the ASD model varied in space/time. This seems a bit speculative (i.e. unsupported).

 Figure 9: This figure is very convincing of the high quality of the OMPS aerosol extinction profile product.

L405: Regarding "corrections", does this need to be plural?

L411: "Angstrom" -> "an Angstrom"

L434: "measurement" -> "magnitude of the measurement"

L439: "in general" -> ", in general,"

L439: is -> are

L441: "rather" -> "a rather"

L445: "different" -> "a different"

L455: shown -> show

L470: rage -> range

L482: Add "in the stratosphere"

L490: Remove "multiple"

---

## Referee Comment (RC2) · Anonymous Referee #2 · 21 Sep 2020

Review of "OMPS LP Version 2.0 Multi-wavelength Aerosol Extinction Coefficient Retrieval Algorithm " by Taha et al.

**General Comments**

This paper describes the version 2 OMPS LP multiwavelength aerosol retrievals. OMPS results are compared to SAGE III, OSIRIS, and CALIPSO, for a variety of altitudes, latitudes, and measurement wavelengths. Sections 1 and 2 are very well written and enjoyable to read.  In sections 3 - 5, however, the writing is of lower quality, with many grammatical errors and poorly formed sentences. The poor writing quality is evident when scanning in Reviewer #1's comments, which also point out many errors. Overall I find the work to be of sufficient quality to warrant publication after some minor revisions as described below. The paper will no doubt be useful to users of the OMPS observations.

**Specific Comments**

1) line 17: Define the acronym PyroCb

2) line 31: "ballon-borne"

3) line 81:  Add "and good vertical resolution"

4) line 129:  Do you mean solar scattering angle (SSA)?

5) line 145:  Here and elsewhere, insert a comma before and after the variable name (e.g. h)

6) line 149: Define the acronyms GSLS RTM

7) line 155: The parenthetical reference should come after the subject

8) line 157: Refractive index can vary with the sulfate composition (wt. % H2SO4), please comment.

9) line 167:  : should be a period, and the parentheses should be deleted.

10) line 208: "line-of-sight"

11) line 211: Please quantify "very small"

12) Figure 1:  The caption does not describe what is in panels a) and b). Please locate "a)" and "b)" before the descriptive text.  Also, consider combining Figures 1 & 2 since the point is to see how things change with scattering angle.

13) line 216:  Please clarify what you term as a cloud, perhaps  "…the cloud layer evident as enhanced extinction near 10.5 km…"

14) line 225 "of" measurements,  also, this is a really long sentence…

15) line 229:  …on the Meteor…

16)  line 235: O4 ?

17) line 238: OSIRIS was already defined above

18) line 261: The lidar ratio can depend on the aerosol size and refractive index (composition), please comment on this.

19) line 263: Please describe the cloud filtering approach in more detail, or add a reference on the method.

20) line 269: This sentence need to be restructured for clarification, also please state the differences (%) with SAGE III/ISS

21) line 281 "1 km vertical intervals"

22) line 290: SAGE II is filtered for what? cirrus? PyroCb's ?

23) line 294 "model" should be "distribution", also, "ASD" was defined above

24) line 300: No need to redefine SSA

25) lines 302-303: It is not clear how the results in Fig. 3 demonstrates that the algorithm is insensitive to errors in the assumed ASD. You need to justify this statement with additional detail.

26) line 310: By reflecting surface, do you mean Earth surface?

27) lines 319-320: this sentence should be clarified.

28) line 321: Remove the parentheses from this sentence.

29) Figure 4: put the letters (e.g., "(a)") before the description.

30) line 337 and elsewhere): The preferred syntax would be "869 nm wavelength"

31) line 336: This paragraph is a bit clumsy overall.

32) line 351: There is no need to list the wavelengths at the end of this sentence.

33) line 352: To be precise, the comparisons do not show this. You deduce this, based on your knowledge of OMPS, and the comparison differences.

34) Figure 8: Please correct the label on the color bar, which should say 1 - the standard deviation of the difference (or 1 - sigma).

35) line 369: This is one example of a poorly formed sentence, which seem to be common in this section. "Based on SAGE III comparison, …" should be "Based on the comparisons with SAGE III,…"

36) line 386: Did OMPS measure "more aerosol" or "report higher extinctions"? Please clarify.

37) line 387: What do you mean by "heavily skewed by few daily measurements…"? Please explain this effect.

38) line 389: "…use of a fixed…"

39) line 393: I do not see how differences in vertical resolution could lead to differences in the time series of extinction after a volcanic eruption. These statements seem misdirected. Please clarify your thoughts on this, and / or consider other explanations.

40) line 400: Please remind us which Figure you are discussing.

41) Figure 12: This is a bit of a challenge to interpret. It might be improved by adding a legend to the figure, and using unique colors.

42) Figures 13 & 14: Referring back to Figure 12 for a description of the lines is tedious, please add captions to the figures.

43) line 429: should be "…18.5 km in the tropics.."; this is just one example of poor grammar in this section.

44) Figure 15: "Top panels show the…". Also, the color scale for panels d - f should indicate the units as (%).

45)

---

## Author Comment (AC1) · 1 Dec 2020

Referee #1

We would like to thank Reviewer 1 for his/her helpful comments which improved the quality of the manuscript. Our responses to the reviewer are listed below, the reviewer comments in italic and our response in regular font.

*I have combined the scientific/major comments with some wording suggestions below. The work is certainly publishable after addressing some of the revisions, but I feel the method can be improved (see below) and wonder if improved results would follow.*

*L166: "$h_n$= the normalization tangent height = 40.5 km" -> "$h_n$, namely the normalization tangent height, set to a value of 40.5 km"*

Done

*L169: Similar -> Analogous*

Done

*L182-185: This is more constrained, not relaxed, on the low end (0.333 vs 0.2). Is "0.2" a typo?*

The 1/3 value is an error, and has been corrected to 0.2 (same as the value used in the V1 algorithm).

*L201: This is my first major criticism: stopping the retrieval based on convergence at one height is a mistake. This will mean poor low accuracy particularly below this tangent height but even at altitudes at/above this tangent height.*

We understand the reviewer's concern and we are already planning on modifying the algorithm to check for convergence for all altitudes of the main aerosol layer, which will be released next year as Version 2.1. However, our analysis indicates that this change will have a limited impact on the retrieved aerosol profiles, mostly affecting the shorter wavelengths (675 nm or less), and only in the tropics in the aftermath of large aerosol enhancements such as volcanic eruptions, when it converges after 3 iterations. We have revised the sentence and it reads as "Iterations end when the retrieved aerosol extinction changes by < 2% at 20 km or when it reaches maximum number of iterations. The planned V2.1 release next year will use modified convergence criteria that checks for multiple altitudes."

*L227: "it" is not defined*

*We replaced "it" with "the retrieval"*

*L235: "by" -> "caused by"*

Done

*L236: Sect 3.1 should state something more about SAGE III cloud detection if SAGE III data have been filtered for clouds in this paper. This becomes relevant with the statement at L395.*

The cloud screening of all instruments is addressed in section 4. We have modified the text in L299 to add more details to the cloud clearing process. The text now reads "SAGE III is filtered for cloud contamination by using only data with extinction ratio at 510 nm / 1022 nm greater than 2 (Thomason and Vernier, 2013)"

*L258: Regarding "retrieve extinction", CALIPSO does not really retrieve extinction.*

We replaced "retrieve" with "obtain"

*L261: retrieve -> obtain'*

Done

*L269: 30km -> 30 km (see also L344, L468)*

Done

*L281: Where -> where*

Done

*L288: What is a "cloud type flag" and how is this determined?*

We have added the following text to explain the cloud types and how it is determined: "Cloud type classifies the identified cloud as cloud, enhanced aerosol, or PSC. The "enhanced aerosol" definition requires the cloud altitude to be at least 1.5 km above the tropopause. The "PSC" definition requires the cloud altitude to be at least 4 km above the tropopause, and the ancillary temperature at the cloud altitude to be less than 200 K."

*L305: "negative 1%" -> "-1% per degree"*

Done

*L305: Why is "Junge" italicized?*

*"Junge" is changed to* "Junge"

*L306: "in the northern hemisphere high latitude"-> "at northern high latitudes"*

Done

*Figure 3: Why is there a wavelength dependence to this error? The wavelength dependence is obvious at 16.5 km. Are all months included here? Without doing any calculations, it seems that there should be some larger scattering angle differences when the sun is low (winter) for 60-70 degrees N. Why is the sign of the SSA dependence of the aerosol extinction difference changing (y-axis) between 25 and 16 km? I wonder if specifying the convergence criterion only at 20 km (see comment above) is partly responsible for the lack of SSA dependence at 20 km, while there is an obvious dependence at 25 and 16.5 km.*

Wavelength dependence of this error is generally expected since the different wavelengths have different sensitivity to aerosol particle size (Reiger et al., 2014). As we explained in the text (L304 – L306), because of the spacecraft orbit, these measurements only take place at high latitudes during the summer of both hemispheres. OMPS LP observations at high latitudes during the winter are mostly in the dark.

We are confident that the convergence criteria have no effect on the analysis shown in Figure 3, since these analyses were made in the NH for a period not affected by any volcanic perturbations (see our comment above). The aerosol differences seen at 16 and 25 km are largely driven the uncertainty in the assumed aerosol model. Lack of SSA dependence at 20 km means that the a-priori aerosol model used in the retrieval is more representative of the measured aerosol at 20 km. Similar pattern, albeit with larger difference, was found in V1.0 retrieval algorithm that used bimodal aerosol size model and different convergence criteria (see section 2.2.1). The retrieved aerosol extinction dependencies on the SSA were subsequently reduced in V1.5, which indicates that the gamma distribution ASD used in V1.5 is more accurate than the bi-modal ASD used in V1.0 (see figure 1 below). Similar pattern was also seen by Rieger et al. (2019).

We have also added the following text "The main assumption is that, if the retrieved aerosol values are different when the instrument is measuring the same air mass but with different scattering angle, then there is an error in the assumed phase function and ASD model. As shown by Rieger et al. (2019), the ASD errors can introduce seasonal variations that correlate well with the SSA." We also added "Similar analyses made by Rieger et al. (2019) have shown that the OSIRS V7.0 aerosol extinction SSA dependence is 0.5% per degree."

[Figure]

Figure 1: (left panel) top row shows daily average solar zenith angles (SZA) and SSA of the ascending and descending measurements from 70° N to 80° N. Red and green are for ascending v1.0, v1.5 respectively, while dark and light blue are for descending v1.0 and v1.5 respectively. The left side of the middle and bottom rows shows the retrieved aerosol extinction at 20.5 and 16.5 km, while the right side is the difference between the two measurements at the same altitudes. Red is v1.0 and green is v1.5.

*L316: Since at L290, the authors inform us that clouds are being removed, any differences in aerosol extinction between low and high R are therefore not expected differences in extinction due to cirrus, but rather point to cirrus being missed by the cloud flagging. I suppose this is difficult to avoid in the tropical tropopause region.*

That is correct. It is either cloud being missed or incomplete cloud clearing.

*L322: Do the authors believe that the larger R (effective scene reflectivity) at higher latitudes is real or an artifact of the retrieval? If it is real, R should have a seasonal dependence, being higher in winter when there is snow covering the land at northern high latitudes.*

That is correct, we do see seasonal dependence of R being higher in the winter, and lower during the summer, although it is nowhere as low in the tropics. We have revised the sentence and it now reads "Outside the tropics, R mean value is generally greater

than 0.3, with strong seasonal dependence that peaks in the winter. Therefore, any observed differences outside the tropics are uncorrelated with cloud presence."

*Figure 4: This figure can be improved since the contours also look like dashed lines in certain spots.*

Done

*L337: wavelength 869 -> 869*

Done

*L343: Delete "wavelengths"*

Done

*L344: Lack of sensitivity does not necessarily result in bias, in should result in larger noise. Can the authors say anything insightful about the bias (e.g. the sign of the bias)?*

We agree with the reviewer that this sentence is not accurate. The accuracy of the 510 nm is discussed in more details in section 5.2, figure 12, which shows OMPS 510 nm exhibiting periodic jumps in the aerosol extinction value. This is caused by the algorithm's reduced accuracy when the measurement vector is very small. We have now replaced this sentence with "This is an artifact in OMPS retrieval algorithm, which often results in noisy and large extinction values when the measurement vector is too small (see Figure 12)."

*Figure 6: It seems a bit odd that the aerosol extinction bias relative to SAGE is higher at 600 nm than 510 nm for low latitudes at/below 18 km?*

This is most likely caused by the ozone contamination for both OMPS and SAGE retrievals at this wavelength.  We have added the following text "This is due to the ozone interreference in both OMPS and SAGE III 600 nm aerosol retrievals."

*Figure 6: The retrieval clearly has a systematic error in the southern hemisphere. The behavior in the northern mid-latitudes is what one would expect and hope for....*

We disagree with the reviewer. While this is true for the shorter wavelengths (675 or less), the longer wavelengths have strong sensitivity to aerosol, even in the SH (see section 2.2.2). Figures 5, 7, and 9, also show the longer wavelengths agreement with SAGE III is mostly within 10% for most altitudes.

*L359: of -> to*

Done

*L369: 10% is a bit optimistic, particularly at 30 km. Could "~10%" be written instead?*

*Done*

*L374: instruments -> instruments'*

Done

*L388: This sentence is unclear. The authors imply that the difference would be even less than 20% if the ASD model varied in space/time. This seems a bit speculative (i.e. unsupported).*

We deleted the sentence.

*Figure 9: This figure is very convincing of the high quality of the OMPS aerosol extinction profile product.*

We are glad that the reviewer shares our assessment of the high quality of the V2.0 OMPS LP aerosol. Figure 9 clearly support the argument made above, that the V2.0 longer wavelengths are of good quality in the SH.

*L405: Regarding "corrections", does this need to be plural?*

We changed it to correction.

*L411: "Angstrom" -> "an Angstrom"*

Done

*L434: "measurement" -> "magnitude of the measurement"*

Done

*L439: "in general" -> ", in general,"*

Done

*L439: is -> are*

Done

*L441: "rather" -> "a rather"*

Done

*L445: "different" -> "a different"*

Done

*L455: shown -> show*

Done

*L470: rage -> range*

Done

*L482: Add "in the stratosphere"*

Done

*L490: Remove "multiple"*

Done

**References**

Rieger, L. A., Bourassa, A. E., and Degenstein, D. A.: Stratospheric aerosol particle size information in Odin-OSIRIS limb scatter spectra, Atmos. Meas. Tech., 7, 507–522, https://doi.org/10.5194/amt-7-507-2014, 2014.

Rieger, L. A., Zawada, D. J., Bourassa, A. E., and Degenstein, D. A.: A multi-wavelength retrieval approach for improved OSIRIS aerosol extinction retrievals, J. Geophys. Res.-Atmos., 124, https://doi.org/10.1029/2018JD029897, 2019.

Thomason, L. W. and Vernier, J.-P.: Improved SAGE II cloud/aerosol categorization and observations of the Asian tropopause aerosol layer: 1989–2005, Atmos. Chem. Phys., 13, 4605–4616, https://doi.org/10.5194/acp-13-4605-2013, 2013.

---

## Author Comment (AC2) · 1 Dec 2020

Referee #2

We would like to thank Reviewer 1 for his/her helpful comments which improved the quality of the manuscript. Our responses to the reviewer are listed below, the reviewer comments in italic and our response in regular font.

*Review of "OMPS LP Version 2.0 Multi-wavelength Aerosol Extinction Coefficient Retrieval Algorithm "by Taha et al. General Comments This paper describes the version 2 OMPS LP multiwavelength aerosol retrievals. OMPS results are compared to SAGE III, OSIRIS, and CALIPSO, for a variety of altitudes, latitudes, and measurement wavelengths. Sections 1 and 2 are very well written and enjoyable to read. Insections3-5, however, the writing is of lower quality, with many grammatical errors and poorly formed sentences. The poor writing quality is evident when scanning in Reviewer #1's comments, which also point out many errors. Overall I find the work to be of sufficient quality to warrant publication after some minor revisions as described below. The paper will no doubt be useful to users of the OMPS observations.*

*Specific Comments*

*1)line 17: Define the acronym PyroCb*

We added "pyrocumulonimbus (PyroCb)"

*2)line 31: "ballon-borne"*

Done.

*3)line 81: Add "and good vertical resolution"*

Done

*4)line 129: Do you mean solar scattering angle (SSA)?*

We mean single scattering angle (SSA). We've modified the text accordingly.

*5)line 145: Here and elsewhere, insert a comma before and after the variable name (e.g. h)*

Done.

*6)line 149: Define the acronyms GSLS RTM*

We replaced 'GSLS RTM' it with 'Gauss- Seidel Limb Scattering (GSLS) radiative transfer model (RTM)'

*7) line 155: The parenthetical reference should come after the subject*

Done

8)line 157: Refractive index can vary with the sulfate composition (wt. % H2SO4), please comment.

We added "(75% $H_2SO_4$)"

9)line 167: :should be a period, and the parentheses should be deleted.

Done.

10)line 208: "line-of-sight"

Done.

11)line 211: Please quantify "very small"

The difference is generally 1 to 2%. We have now added the following text "Recent calculations performed for a RTM comparison project (Zawada et al., 2020) allow the $\rho_c(\lambda,h)$ values computed by the scalar and vector versions of GSLS to be compared, for a variety of atmospheres and illumination conditions.  For the relevant wavelengths (500 nm and greater), these values agree to within 1% or better at 20 km, and within 2% or better at 10 km."

12)Figure 1: The caption does not describe what is in panels a) and b). Please locate "a) "and "b)"before the descriptive text. Also, consider combining Figures 1 & 2 since the point is to see how things change with scattering angle.

We have now fixed the figure's caption.

13) line 216: Please clarify what you term as a cloud, perhaps"...the cloud layer evident as enhanced extinction near 10.5 km..."

We changed the sentence as suggested.

14)line 225 "of" measurements, also, this is a really long sentence...

Done. We also broke the sentence into two.

15)line 229: ...on the Meteor...

Done.

16)line 235: O4 ?

We added the following "oxygen dimer ($O_4$)"

*17)line 238: OSIRIS was already defined above*

We have now removed OSIRIS definition.

*18)line 261:The lidar ratio can depend on the aerosol size and refractive index (composition), please comment on this.*

We rewrote the sentence to "A constant lidar ratio (extinction to backscatter ratio) of 50 sr was used to obtain the extinction profiles, which is a typical value used for stratospheric aerosol background conditions (Kremser et al., 2016)."

*19)line 263: Please describe the cloud filtering approach in more detail, or add a reference on the method.*

We added the reference (Kar et al., 2019).

*20)line 269: This sentence need to be restructured for clarification, also please state the differences (%) with SAGE III/ISS*

We have revised the sentence and it reads as "However, the difference with SAGE III/ISS at the middle to high latitudes and low altitudes was substantially large, often exceeding 100%."

*21)line 281 "1 km vertical intervals"*

Done.

*22)line 290: SAGE II is filtered for what? cirrus? PyroCb's ?*

We have modified the sentence and it reads as "SAGE III is filtered for cloud contamination by using only data with extinction ratio at 510 nm / 1022 nm greater than 2 (Thomason and Vernier, 2013)"

*23)line 294 "model" should be "distribution", also, "ASD" was defined above*
Done.

*24)line 300: No need to redefine SSA*

We deleted the SSA definition.

*25)lines 302-303: It is not clear how the results in Fig. 3 demonstrates that the algorithm is insensitive to errors in the assumed ASD. You need to justify this statement with additional detail.*

We never made the claim that the algorithm is insensitive to the errors of the assumed aerosol model. The main assumption here is that, if the retrieved aerosol values are different when the instrument is measuring the same air mass but with different scattering angle, then there is an error in the assumed phase function and ASD model. As shown by Rieger et al. (2019), the ASD errors can introduce seasonal variations that correlates well with the SSA. Figure 3 shows little dependency on the SSA at 20 km, and somehow larger dependency at 16 and 25 km. Those results are similar or better than the 0.5% per degree reported by Rieger et al. (2019).

We have modified the text to include the following sentences "The main assumption is that, if the retrieved aerosol values are different when the instrument is measuring the same air mass but with different scattering angle, then there is an error in the assumed phase function and ASD model. As shown by Rieger et al. (2019), the ASD errors can introduce seasonal variations that correlates well with the SSA." And "Similar analyses made by Rieger et al. (2019) has shown that the OSIRS V7.0 aerosol extinction SSA dependence is 0.5% per degree."

*26)line 310: By reflecting surface, do you mean Earth surface?*

Not necessary, it can also be clouds or aerosols. We have now added the following sentence "It doesn't mean the Earth's surface reflectivity, since the scene can contain clouds or aerosols."

*27)lines 319-320: this sentence should be clarified.*

We have revised the sentence and it now reads "Outside the tropics, *R* mean value is generally greater than 0.3, with strong seasonal dependence that peaks in the winter. Therefore, any observed differences outside the tropics are uncorrelated with cloud presence."

*28)line 321: Remove the parentheses from this sentence.*

Done.

*29)Figure 4: put the letters (e.g., "(a)") before the description.*

Done.

*30)line 337and elsewhere): The preferred syntax would be "869 nm wavelength"*

We deleted "wavelength" as suggested by the reviewer 1.

*31)line 336: This paragraph is a bit clumsy overall.*

We have revised the paragraph and it now reads as
"Figure 6 is a summary plot of the mean difference between OMPS and SAGE III coincidences for wavelengths 510, 600, 675, 745, 869, and 997 nm. In general, 869 nm

is the best OMPS retrieved wavelength relative to SAGE III with differences of 5% or less for most altitudes and latitudes. Other wavelengths agree with SAGE III to within 10%. Exceptions to this occur at high altitudes (above ~28 km) where the aerosol loading is minimal, and near the tropopause, which is affected by cloud contamination. The 510 and 600 nm OMPS extinction values have a slightly larger bias of 20% in the tropics. This is due to the ozone interference in both OMPS and SAGE III 600 nm aerosol retrievals. The 997 nm OMPS extinction values have systematic bias of -10% between 60°S and 20°N, caused by stray light contamination in the OMPS measurements. Unlike the other wavelengths, the 997 nm laboratory characterization is poor, and its stray light correction therefore has lower quality (Jaross et al., 2014). In the SH, 510 nm shows large positive bias relative to SAGE III below 18 km. This is an artifact in OMPS retrieval algorithm, which often results in noisy and large extinction values when the measurement vector is too small, (see Figure 12)."

*32)line 351: There is no need to list the wavelengths at the end of this sentence.*

Done.

*33)line 352:To be precise, the comparisons do not show this. You deduce this, based on your knowledge of OMPS, and the comparison differences.*

We deleted "The comparison shows that".

*34)Figure 8: Please correct the label on the color bar, which should say 1 -the standard deviation of the difference (or 1 -sigma).*

Done.

*35)line 369: This is one xample of a poorly formed sentence, which seem to be common in this section. "Based on SAGE III comparison, ..."should be "Based on the comparisons with SAGE III,..."*

We changed the sentence to start with "Based on the comparison with SAGE III,"

*36)line 386: Did OMPS measure "more aerosol "or "report higher extinctions"? Please clarify.*

We have changed the sentence to "where OMPS LP initially reported higher aerosol extinction than SAGE III."

*37)line 387: What do you mean by "heavily skewed by few daily measurements..."? Please explain this effect.*

We agree that this sentence is not clear, so we deleted it. The original text that reads "This might be caused by the different coverage and frequency of measurements for each instrument." Is sufficient enough to explain the differences.

*38)line 389: "...use of a fixed..."*

We deleted this sentence in response to reviewer 1.

*39)line 393: I do not see how differences in vertical resolution could lead to differences in the time series of extinction after a volcanic eruption. These statements seem misdirected. Please clarify your thoughts on this, and / or consider other explanations.*

Vertical resolution differences were previously reported by various studies (Chen et al., 2020; Bourassa et al., 2019).
We have now added the following text "Bourassa et al. (2019) compared nearby OMPS LP and SAGE III/ISS aerosol profiles following the aftermath of the British Columbia fires in 2017. They showed that both instruments have very similar layered vertical structure and magnitude. However, they noted that some differences in layer height and magnitude can be expected from differing vertical resolutions."

*40)line 400: Please remind us which Figure you are discussing.*

We added "(Figure 11)" to the text

*41)Figure 12: This is a bit of a challenge to interpret. It might be improved by adding a legend to the figure, and using unique colors.*

Done.

*42)Figures 13 & 14: Referring back to Figure12 for a description of the lines is tedious, please add captions to the figures.*

Done.

*43)line 429: should be "...18.5 km in the tropics.."; this is just one example of poor grammar in this section.*

Done.

*44)Figure 15: "Top panels show the...". Also, the color scale for panels d -f should indicate the units as (%).*

Done.

**References**

Bourassa, A., Rieger, L., Zawada, D. J., Khaykin, S., Thomason, L., and Degenstein, D.: Satellite limb observations of unprecedented forest fire aerosol in the stratosphere, J. Geophys. Res., 124, 9510-9519, https://doi.org/10.1029/2019JD030607, 2019.

Chen, Z., Bhartia, P. K., Torres, O., Jaross, G., Loughman, R., DeLand, M., Colarco, P., Damadeo, R., and Taha, G.: Evaluation of the OMPS/LP stratospheric aerosol extinction product using SAGE III/ISS observations, Atmos. Meas. Tech., 13, 3471–3485, https://doi.org/10.5194/amt-13-3471-2020, 2020.

Kar, J., Lee, K.-P., Vaughan, M. A., Tackett, J. L., Trepte, C. R., Winker, D. M., Lucker, P. L., and Getzewich, B. J.: CALIPSO level 3 stratospheric aerosol profile product: version 1.00 algorithm description and initial assessment, Atmos. Meas. Tech., 12, 6173–6191, https://doi.org/10.5194/amt-12-6173-2019, 2019.

Kremser, S., Thomason, L.W., Hobe, M., Hermann, M., Deshler, T., Timmreck, C., Toohey, M., Stenke, A., Schwarz, J. P.,Weigel, R., Fueglistaler, S., Prata, F. J., Vernier, J.-P., Schlager, H., Barnes, J. E., Antu na-Marrero, J.-C., Fairlie, D., Palm, M., Mahieu, E., Notholt, J., Rex, M., Bingen, C., Vanhellemont, F., Bourassa, A.,Plane, J. M. C., Klocke, D., Carn, S. A., Clarisse, L., Trickl, T., Neely, R., James, A. D., Rieger, L., Wilson, J. C., and Meland, B.: Stratospheric aerosol-Observations, processes, and impact on climate, Rev. Geophys., 54, 278–335, 2016.

Rieger, L. A., Zawada, D. J., Bourassa, A. E., and Degenstein, D. A.: A multi-wavelength retrieval approach for improved OSIRIS aerosol extinction retrievals, J. Geophys. Res.-Atmos., 124, https://doi.org/10.1029/2018JD029897, 2019.

Thomason, L. W. and Vernier, J.-P.: Improved SAGE II cloud/aerosol categorization and observations of the Asian tropopause aerosol layer: 1989–2005, Atmos. Chem. Phys., 13, 4605–4616, https://doi.org/10.5194/acp-13-4605-2013, 2013.

---

## Author Response (AR1)

**OMPS LP Version 2.0 Multi-wavelength Aerosol Extinction Coefficient Retrieval Algorithm**

Ghassan Taha1,2, Robert Loughman3, Tong Zhu4, Larry Thomason5, Jayanta Kar6Kar5,6, Landon Rieger7, and Adam Bourassa7

1Universities Space Research Association (USRA), Greenbelt, Maryland, USA
 2NASA Goddard Space Flight Center, Greenbelt, Maryland, USA

[revised manuscript text omitted]

---

## Referee Report (RR1)

Re-review of Taha et al. (submitted to AMT)

I congratulate the authors on improving the manuscript point-by-point. I am a bit surprised, with retrieval convergence only forced at 20 km, that the algorithm does so well at capturing the temporal and latitudinal variations at 25.5 km. While I believe this algorithm could easily be improved in this regard, the results speak for themselves. Here are some minor points:

L235: It is not physically correct to refer to $O_2$-$O_2$ collision complex as the $O_4$ dimer. $O_4$ dimers only exist at pressures higher than found in Earth's atmosphere (~1 atm). I was OK with $O_4$ since, for a split second, it is a complex containing 4 oxygen atoms, but I am not OK with dimer.

L355: The positive bias for 510 nm aerosol extinction remains an unsolved mystery. I am fine with the hope that this may be resolved in the future. It is difficult to measure aerosol extinction at such a short wavelength from my experience because its contribution to the radiance becomes small relative to other variables such as ozone, and (pressure-dependent) Rayleigh scattering.

Figure 6: Just to clarify my comment from review: the differences are not symmetric about the 0% line in the southern mid-latitudes, whereas in the tropics, there is a wavelength-dependent bias with little altitude dependence and in the northern mid-latitudes, the results are very encouraging below 24 km with a wavelength-dependent bias only above that altitude.

---

## Author Response (AR2)

Referee #1

We would like to thank Reviewer 1 for his helpful comments. Our responses to the reviewer are listed below, the reviewer comments in italic and our response in regular font.

*I congratulate the authors on improving the manuscript point-by-point. I am a bit surprised, with retrieval convergence only forced at 20 km, that the algorithm does so well at capturing the temporal and latitudinal variations at 25.5 km. While I believe this algorithm could easily be improved in this regard, the results speak for themselves. Here are some minor points:*

*L235: It is not physically correct to refer to O2-O2 collision complex as the O4 dimer. O4 dimers only exist at pressures higher than found in Earth's atmosphere (~1 atm). I was OK with O4 since, for a split second, it is a complex containing 4 oxygen atoms, but I am not OK with dimer.*

The term "$O_4$ dimer" was referred to as such in (Wang et al., 2020) SAGE III/ISS ozone validation paper. However, we have now deleted "oxygen dimer", and we used "O4 absorption cross section" as referred to by Thomason et al., 2020.

*L355: The positive bias for 510 nm aerosol extinction remains an unsolved mystery. I am fine with the hope that this may be resolved in the future. It is difficult to measure aerosol extinction at such a short wavelength from my experience because its contribution to the radiance becomes small relative to other variables such as ozone, and (pressure-dependent) Rayleigh scattering.*

We have seen similar large values at 675 nm when the measurement vector (ASI) is very low. However, most of those large values were filtered out using a threshold of ASI value less than 0.01. As the referee pointed out, it is difficult to retrieve the aerosol at 510 nm low altitudes when the aerosol signal is very small relative to ozone absorption and Rayleigh scattering. We hope to better filter out most of those erroneous retrievals in a future release. We have revised the sentence to "This is an artifact in the OMPS retrieval algorithm, which often results in noisy and large extinction values when the measurement vector is too small relative to gaseous absorption and Rayleigh scattering (see Figure 12)."

*Figure 6: Just to clarify my comment from review: the differences are not symmetric about the 0% line in the southern mid-latitudes, whereas in the tropics, there is a wavelength-dependent bias with little altitude dependence and in the northern mid-latitudes, the results are very encouraging below 24 km with a wavelength-dependent bias only above that altitude.*

Thanks for the explanation. While we agree with the reviewer, figures 5, 7, and 9 show the longer wavelengths agreement with SAGE III is mostly within 10% for most altitudes and latitudes. Furthermore, some of the bias seen at the tropics and southern mid-latitude below 18 km is caused by the incomplete cloud clearing which might explain the non-symmetric differences.

Thomason, L. W., Kovilakam, M., Schmidt, A., von Savigny, C., Knepp, T., and Rieger, L.: Evidence for the predictability of changes in the stratospheric aerosol size following volcanic eruptions of diverse magnitudes using space-based instruments, Atmos. Chem. Phys. Discuss., https://doi.org/10.5194/acp-2020-480, in review, 2020.

Wang, H., Damadeo, R., Flittner, D., Kramarova, N., Taha, G., Davis, S., Thompson, A., Strahan, S., Wang, Y., Froidevaux, L., Degenstein, D., Bourassa, A., Steinbrecht, W., Walker, K., Querel, R., Leblanc, T., Godin-Beekmann, S., Hurst, D., Hall, E.: Validation of SAGE III/ISS Solar Ozone Data with Correlative Satellite and Ground Based Measurements, J. Geophys. Res.-Atmos, https://doi.org/10.1029/2020JD032430, 2020.